

# Bayesian inference of earthquake rupture models using polynomial chaos expansion

Hugo Cruz-Jiménez[1], Guotu Li[2], Paul Martin Mai[1], Ibrahim Hoteit[1], and Omar M. Knio[1,2]

[1]King Abdullah University of Science and Technology, Thuwal 23955, Saudi Arabia
[2]Duke University, Durham, NC 27708, USA

*Correspondence to:* Guotu Li (guotu.li@duke.edu); Omar M. Knio (Omar.Knio@kaust.edu.sa);

**Abstract.** In this paper we employed polynomial chaos (PC) expansions to understand earthquake rupture model responses to random fault plane properties. A sensitivity analysis based on our PC surrogate model suggests that the hypocenter location plays a dominant role in peak ground velocity (PGV) responses, while elliptical patch properties only show secondary impact. In addition, the PC surrogate model is utilized for Bayesian inference of the most likely underlying fault plane configuration

5 in light of a set of PGV observations from a ground motion prediction equation (GMPE). A restricted sampling approach is also developed to incorporate additional physical constraints on the fault plane configuration, and to increase the sampling efficiency.

**Keywords.** Polynomial Chaos expansion, Sensitivity analysis, Bayesian inference, Earthquake seismology, Peak ground velocity.

## 1 Introduction

One of the most important challenges seismologists and earthquake engineers face to design large civil structures (e.g. buildings, dams, bridges, power plants) and response plans, especially in highly populated cities prone to large damaging earthquakes, is the reliable estimation of ground-motion characteristics at a given location. Ground-motion prediction equations

15 (GMPEs), which are one of the most important elements for Probabilistic Seismic Hazard Analysis (PSHA), are designed for this purpose. These are obtained from regression analysis by fitting a dataset (empirical and simulated) and are mainly expressed in terms of the site conditions, source-site distance (e.g. rupture distance or Joyner-Boore distance, denoted as $R_{JB}$ distance hereafter[1]), magnitude and mechanism, although other terms such as directivity and hanging wall effect are also considered (Abrahamson et al., 2014). The equations can be derived for peak ground displacement (PGD), peak ground velocity

20 (PGV), peak ground acceleration (PGA), and spectral acceleration (SA) for a damping of 5% at different periods. Ideally, an optimal GMPE has to be robust, and include physical terms to avoid over fitting the data, which can result in the inclusion

---

[1]The Joyner-Boore distance is defined as the shortest distance from a site to the surface projection of the rupture plane.





of too many parameters. When other effects are considered (such as amplitude and duration of rupture directivity (Somerville et al., 1997)) or more data is available (Atkinson and Boore, 2011), GMPEs are modified to better explain attenuation patterns.

Many efforts have been made to characterize the seismic ground-motion considering both real and simulated data. For example, using real data, five research groups under the Pacific Earthquake Engineering Research Center Next Generation Attenuation (PEER NGA) project derived GMPEs for shallow crustal earthquakes considering an extensive database of recorded ground-motions (Chiou et al., 2008). Later, Arroyo and Ordaz (2010a, b) obtained GMPEs using both synthetic data and two subsets of accelerograms of the NGA database (Chiou et al., 2008). Arroyo and Ordaz (2010b) highlighted the necessity to merge finite fault modeling (Atkinson and Silva, 2000) with observations to obtain GMPEs that better predict the amplitudes in zones where data is insufficient. Verification and validation studies (Maufroy et al., 2015, 2016) were also conducted in a large effort to understand ground motions and showed the importance of both accurate source parameters and the geological description of the medium to reproduce observed ground motions. Singh et al. (2017) improved the agreement between observed ground motions and GMPEs by including site effects of the area. Numerical simulations have also helped to explain ground-motion characteristics. For instance, Furumura and Singh (2002) described attenuation patterns for both deep in-slab and shallow interplate earthquakes, while Cruz-Jiménez et al. (2009) explained ground-motion amplification due to a volcanic layer. Mahani and Atkinson (2012) modeled the decay of spectral amplitudes in several locations in North America.

In this study we investigate the level of complexity needed in kinematic rupture models of Mw 6.5 strike-slip events to produce ground-motions similar to a reference GMPE. To this end, we utilize the PC approach (Ghanem and Spanos, 1991; Xiu and Karniadakis, 2002; Le Maître and Knio, 2010) to build functional representations of PGVs responses of an original source model. Thanks to the significant reduction in computational cost of the PC surrogate models (in comparison with both the original source model and a Bayesian analysis based on MCMC sampling, which requires a prohibitive number of model runs (Minson et al., 2014)), it is suitable to utilize the PC surrogates in a Bayesian inference framework (Sudret and Mai, 2013; Sraj et al., 2016; Giraldi et al., 2017). This enable us to quantitatively rank different kinematic source models given by the PGVs they produce and identify the most likely one that fits a chosen reference GMPE (expectation). The ranking considers uncertainties in both the GMPE and model parameters. This provides useful insight on the level of complexity needed in kinematic source models for ground-motion simulations to satisfy both observational constraints and engineering/design requirements for seismic safety.

This paper is organized as follows. In Section 2 we provide detailed descriptions of the source model configurations, including the calculation of synthetic seismograms. In section 3, we present the PC analysis of PGVs as a function of random variations of the kinematic models, including the validation of PC surrogate models and discussions of various statistical quantities. In section 4, we conduct a PC based Bayesian inference analysis to identify the most likely kinematic rupture model that best fits a chosen GMPE reference curve. Finally, we conclude our key findings and propose potential improvements for future work in section 5.





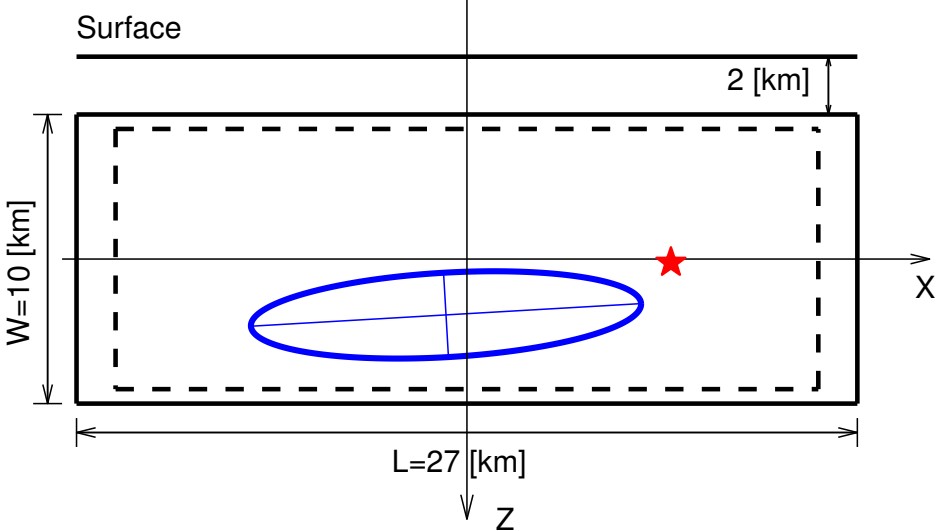

**Figure 1.** Example of fault plane configuration, the red star denotes hypocenter location, and the ellipse is the asperity with Gaussian slip distribution inside. The slip distribution is tapered in the area between the dashed and solid rectangles.

## 2   Source Model

A magnitude $M_w = 6.5$ strike-slip earthquake (seismic moment $6.31 \times 10^{18}$ Nm) on a single-segment vertical fault plane is considered. The fault plane is chosen to be a rectangle with fixed length $L = 27$ km and width $W = 10$ km (obtained from 100 realizations following the scaling relation in Wells and Coppersmith (1994)). The top of the fault plane is located 2 km below the ground surface. Figure 1 shows an example configuration of the fault plane, in which the red star denotes the hypocenter

and the ellipse is the asperity with Gaussian slip distribution inside. The maximum slip $S_{max}$ is chosen such that the mean slip remains constant (0.71 m) when varying the ellipse size. The slip between the elliptical patch boundary and dashed rectangle (Figure 1) is set to be $S_{max}/e$ (where $e$ is the Euler's number). The slip between the solid and dashed rectangles (the horizontal and vertical gaps are 5% of the length and width of the fault plane, respectively) is tapered to avoid non-physical slip patterns. The entire fault plane is discretized in along-strike and down-dip directions with grid size of 0.02 km. We use a regularized

Yoffe function (Tinti et al., 2005) with a rise time $Tr = 1.25$ s following source-scaling relations (Somerville et al., 1999) and slip acceleration time $t_{acc} = 0.225$ s, as suggested by Tinti et al. (2005). At each node of the discretized fault plane we assign $Tr$, $t_{acc}$, slip-rate in along-strike and down-dip directions, and rupture time. We consider a rupture speed of $0.75V_s$ km/s in all source models.

PGVs at a virtual network of $N_{obs} = 56$ stations (Figure 2) are calculated from synthetic seismograms of the two horizontal

components of ground motion at each site for a large set of source rupture models. We use COMPSYN (Spudich and Xu, 2003), a code based on the discrete wavenumber/finite element method proposed by Olson et al. (1984) to calculate the synthetic seismograms up to a maximum frequency of 1.5 Hz at each station of the virtual array. This approach considers a layered





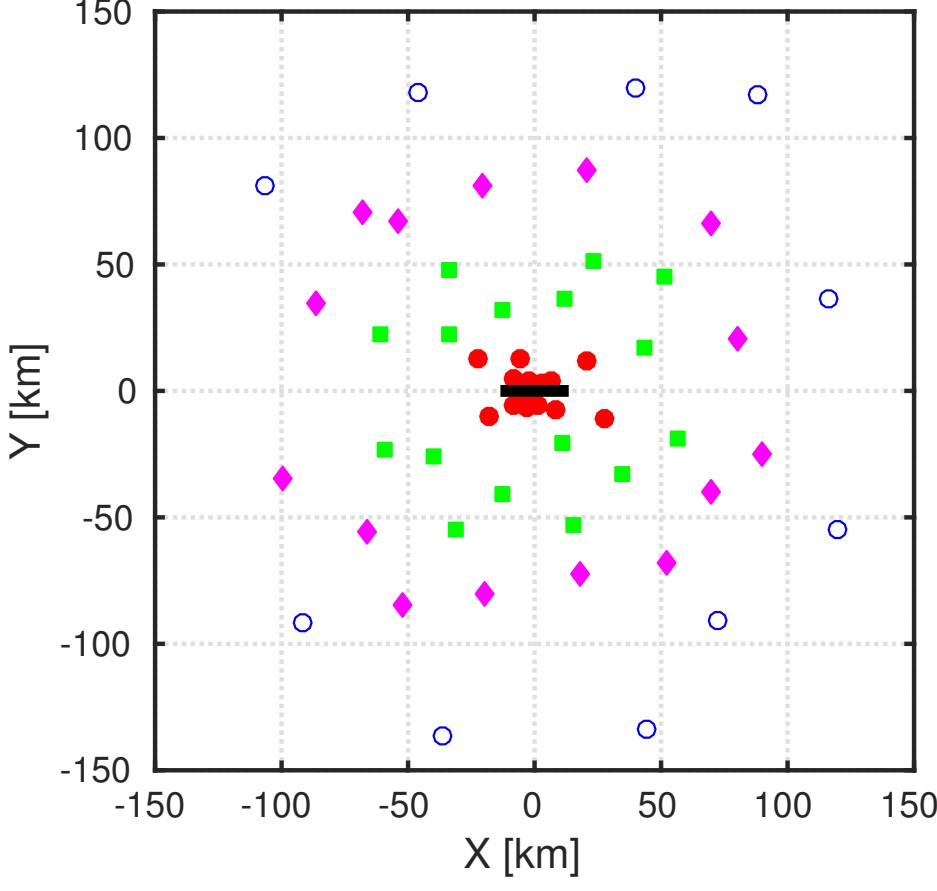

**Figure 2.** A virtual network of $N_{obs} = 56$ stations where PGV responses are reported by the source model. The solid black line at the center denotes the length and location of the fault plane.

**Table 1.** Velocity model used in this study, modified from Boore et al. (1997).

| Depth (km) | $V_p$ (km/s) | $V_s$ (km/s) |
|---|---|---|
| 0 | 2.4 | 1.5 |
| 0.5 | 4.4 | 2 |
| 1.5 | 5.3 | 2.7 |
| 2.5 | 5.5 | 2.9 |
| 4 | 5.7 | 3.3 |
| 8 | 6.1 | 3.5 |
| 14 | 6.8 | 3.9 |
| 16.6 | 7.1 | 4.1 |
| 27 | 8 | 4.6 |
| 350 | 8.2 | 4.65 |





**Table 2.** Parameters governing fault plane configurations, (*) denotes dependent parameters.

| Index | Parameter | Physical Interpretation |
|-------|-----------|-------------------------|
| 1 | $AR$ | Area ratio, $AR = \frac{\pi ab}{L*W} \in [0.05, 0.29]$ |
| 2 | $x_h \, (km)$ | x-coordinate of the hypocenter $x_h \in [-13.5, 13.5]$ |
| 3 | $z_h \, (km)$ | z-coordinate of the hypocenter $y_h \in [-5, 5]$ |
| 4 | $a \, (*) \, (km)$ | Semi-major axis $a \in [\sqrt{\frac{AR \cdot L \cdot W}{\pi}}, L/2]$ |
| 5 | $\theta \, (*)$ | Inclination angle of the elliptical patch |
| 6 | $x_c \, (*) \, (km)$ | x-coordinate of the center of elliptical patch |
| 7 | $z_c \, (*) \, (km)$ | z-coordinate of the center of elliptical patch |

1D velocity structure. We apply the velocity model shown in Table 1, which corresponds to a slightly modified version of the generic model by Boore et al. (1997) for California. PGVs serve as our quantities of interest (QoIs, each denoted as $\mathcal{Q}_j$, for $j = 1, 2, ..., N_{obs}$). We aim at understanding stochastic source model PGV responses to random fault plane configurations of the source process (slip distributions and hypocenter location). To this end, we consider variations in seven physical parameters listed in Table 2, which parameterize the fault plane configurations, i.e. locations of both the hypocenter and elliptical asperity patch, as well as its shape and orientation. We restrict the hypocenter and elliptical patch to be inside the fault plane, and limit

the area aspect ratio (AR) of the elliptical patch to the entire fault plane ($L \times W$) between 5% and 29%. These restrictions lead to nonlinear dependency between feasible ranges of different physical parameters (see Appendix A for more details).

## 3 Polynomial Chaos Framework

PC expansions (Ghanem and Spanos, 1991; Xiu and Karniadakis, 2002; Le Maître and Knio, 2010) are used in this study to understand earthquake rupture model responses (in terms of PGVs) to random configurations of slip distribution and hypocen-

10 ter location. We associate each of the physical parameters with a random variable $\xi_i$ ($i \in \{1, 2, ..., n_d\}$, where $n_d = 7$ is the number of stochastic dimensions) and assume all $\xi_i$'s are independent and uniformly distributed over $[-1, 1]$. That is, the joint distribution of the random parameter vector $\boldsymbol{\xi}$ is

$$p(\boldsymbol{\xi}) = \begin{cases} 2^{-7} & \text{if } \boldsymbol{\xi} \in \Xi \equiv [-1, 1]^7, \\ 0 & \text{otherwise.} \end{cases} \tag{1}$$

Each random parameter vector $\boldsymbol{\xi} \in \Xi$ can be linked uniquely to a realization of the physical parameter vector (See mapping

details in Appendix A). We thus focus on constructing functional representations of PGV responses at each station with respect to $\boldsymbol{\xi}$, instead of the physical parameters in Table 2.



Let $\mathcal{Q}_j(\boldsymbol{\xi})$ be the PGV response to $\boldsymbol{\xi}$ at the $j$-th station ($j \in \{1, 2, ..., N_{obs}\}$), and assume each $\mathcal{Q}_j$ is a second-order random variable, i.e. $\mathcal{Q}_j(\boldsymbol{\xi})$ is in the Hilbert space $L_2(\boldsymbol{\Xi}, p)$ and

$$\mathbb{E}\left[\mathcal{Q}_j^2\right] = \int_{\boldsymbol{\Xi}} \mathcal{Q}_j(\boldsymbol{\xi})^2 p(\boldsymbol{\xi}) d\boldsymbol{\xi} < +\infty, \quad \forall j \in \{1, 2, ..., N_{obs}\}. \tag{2}$$

One can approximate $\mathcal{Q}_j(\boldsymbol{\xi})$ using a truncated PC expansion as follows:

$$\mathcal{Q}_j(\boldsymbol{\xi}) \approx \tilde{\mathcal{Q}}_j(\boldsymbol{\xi}) = \sum_{\alpha=0}^{N_p} c_\alpha \Psi_\alpha(\boldsymbol{\xi}), \quad \forall j \in \{1, 2, ..., N_{obs}\}. \tag{3}$$

where $N_p$ is a truncation parameter and $(N_p + 1)$ is the number of expansion terms retained in the PC surrogate models. In this

study, we truncated the PC expansion at total polynomial order of nine, which leads to 11440 polynomials. By adopting the classical convetion of $\Psi_0(\boldsymbol{\xi}) = 1$, the mean and variance of a PC surrogate $\mathcal{Q}_j(\boldsymbol{\xi})$ can be expressed as:

$$\mathbb{E}\left[\tilde{\mathcal{Q}}\right] = \sum_{\alpha=0}^{N_p} c_\alpha \langle \Psi_\alpha, 1 \rangle = c_0, \tag{4}$$

and

$$\mathbb{V}\left[\tilde{\mathcal{Q}}\right] = \mathbb{E}\left[(\tilde{\mathcal{Q}} - \mathbb{E}[\tilde{\mathcal{Q}}])^2\right] = \sum_{\alpha,\beta=1}^{N_p} c_\alpha c_\beta \langle \Psi_\alpha, \Psi_\beta \rangle = \sum_{\alpha=1}^{N_p} c_\alpha^2 \|\Psi_\alpha\|_{L_2}^2, \tag{5}$$

where $\langle \cdot \rangle$ denotes the inner product in the Hilbert space $L_2(\boldsymbol{\Xi}, p)$ with respect to the joint distribution $p(\boldsymbol{\xi})$ (Le Maître and Knio, 2010).

To determine the expansion coefficients ($c_\alpha$'s) in Eq. (3), we rely on a Latin Hypercube Sample (LHS) (McKay et al., 1979) set (denoted as $\mathcal{P}_{LHS}$ hereafter) of $N_{LHS} = 8000$ earthquake rupture model realizations and solve the following Basis Pursuit Denoising (BPDN) problem (Van Den Berg and Friedlander, 2007, 2008) at each station:

$$\boldsymbol{c}^* = \arg \min_{\boldsymbol{c} \in \mathbb{R}^{N_p+1}} \|\boldsymbol{c}\|_{l_1} \text{ s.t. } \|\boldsymbol{\mathcal{Q}}_j - [\Psi]\boldsymbol{c}\| \le \gamma \|\boldsymbol{\mathcal{Q}}_j\|_{l_2}, \quad \forall j \in \{1, 2, ..., N_{obs}\}, \tag{6}$$

where $\boldsymbol{\mathcal{Q}}_j = (\mathcal{Q}_j(\boldsymbol{\xi}_1), \mathcal{Q}_j(\boldsymbol{\xi}_2), ..., \mathcal{Q}_j(\boldsymbol{\xi}_{N_{LHS}}))^T$ is the model PGV realization vector at the $j$-th station, and $\boldsymbol{c} \in \mathbb{R}^{N_p+1}$ is the coefficient vector for the corresponding PC surrogate model. $[\Psi] \in \mathbb{R}^{N_{LHS} \times (N_p+1)}$ denotes the polynomial matrix with each element $[\Psi]_{i,\alpha} = \Psi_\alpha(\boldsymbol{\xi}_i)$. Note $[\Psi]$ is station invariant. The scalar parameter $\gamma$ indicates the model noise level and is determined numerically via a cross-validation process.

Following Sobol (1993), Homma and Saltelli (1996), variance-based first-order and total order sensitivity indices associated with a subset of random variables ($\boldsymbol{i} \subset \{1, 2, ..., n_d\}$) can be calculated respectively as follows:

$$\mathbb{S}_{\boldsymbol{i}} = \frac{\sum_{\alpha \in \mathcal{S}_{\boldsymbol{i}}} c_\alpha^2 \|\Psi_\alpha\|_{L_2}^2}{\sum_{\alpha=1}^{N_p} c_\alpha^2 \|\Psi_\alpha\|_{L_2}^2}. \tag{7a}$$

$$\mathbb{T}_{\boldsymbol{i}} = \frac{\sum_{\alpha \in \mathcal{T}_{\boldsymbol{i}}} c_\alpha^2 \|\Psi_\alpha\|_{L_2}^2}{\sum_{\alpha=1}^{N_p} c_\alpha^2 \|\Psi_\alpha\|_{L_2}^2}, \tag{7b}$$





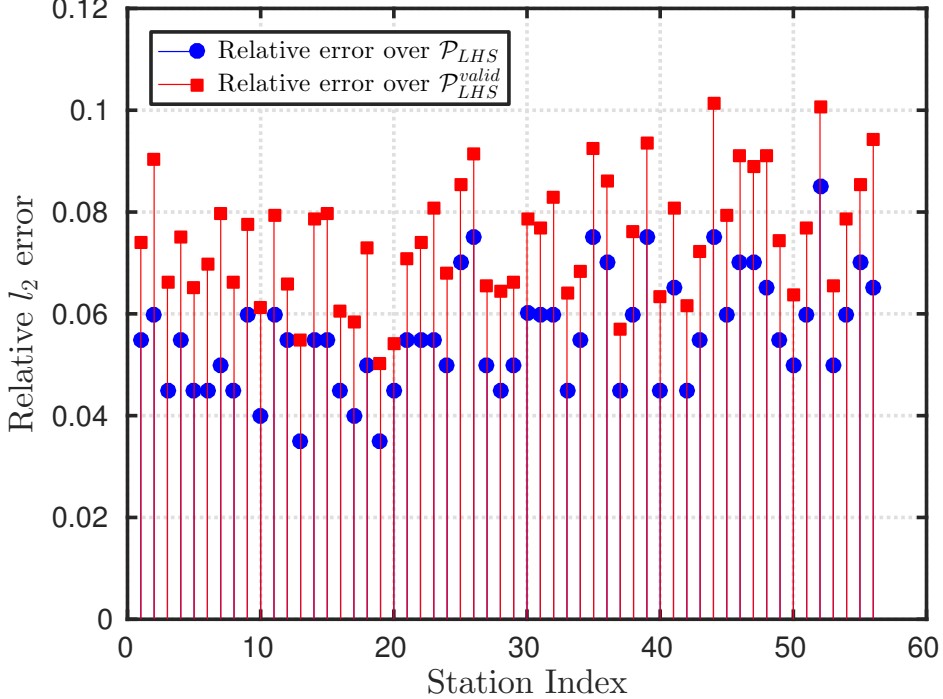

**Figure 3.** Relative $l_2$ errors of PC surrogate models.

where $\mathbb{S}_{\boldsymbol{i}}$ (first-order sensitivity) is the relative variance contribution of those polynomials exclusively related to random vari-
ables in the subset $\boldsymbol{i}$; while $\mathbb{T}_{\boldsymbol{i}}$ (total order sensitivity) is the relative variance contribution of polynomials involving any of the
random variables in $\boldsymbol{i}$ (including cross polynomials between variables in $\boldsymbol{i}$ and its complement $\boldsymbol{i}_{\sim}$, $\boldsymbol{i} \cup \boldsymbol{i}_{\sim} = \{1, 2, ..., n_d\}$). Note
that by definition the two polynomial index sets satisfy $\mathcal{S}_{\boldsymbol{i}} \subset \mathcal{T}_{\boldsymbol{i}}$.

## 3.1  Validation of PC Models

We first validate our PC surrogate models for PGVs at all stations. To this end, we introduce a second independent source
model simulation ensemble (again an 8000 member LHS set $\mathcal{P}_{LHS}^{valid} \subset \Xi$) for the purpose of validation. (Note that $\mathcal{P}_{LHS}^{valid}$ is
5    independent of the training set $\mathcal{P}_{LHS}$). The following relative $l_2$ error is then examined for PGVs at each station.

$$\epsilon_j = \sqrt{\frac{\sum_{k=1}^{N_{LHS}}(\tilde{\mathcal{Q}}_j(\boldsymbol{\xi}_k) - \mathcal{Q}_j(\boldsymbol{\xi}_k))^2}{\sum_{k=1}^{N_{LHS}} \mathcal{Q}_j(\boldsymbol{\xi}_k)^2}}, \ \forall j \in \{1, 2, ..., N_{obs}\}, \tag{8}$$

where $\tilde{\mathcal{Q}}_j(\boldsymbol{\xi}_k)$ and $\mathcal{Q}_j(\boldsymbol{\xi}_k)$ denote PC and source model responses, respectively, to $\boldsymbol{\xi}_k$ at the $j$-th station. $\boldsymbol{\xi}_k \in \mathcal{P}_{LHS}$ or
$\boldsymbol{\xi}_k \in \mathcal{P}_{LHS}^{valid}$ depending on the sample set used to estimate the errors.

Figure 3 shows relative error estimates of PC surrogate models over the training set ($\mathcal{P}_{LHS}$, blue dots) and the validation
10    set ($\mathcal{P}_{LHS}^{valid}$, red dots). It is not surprising to see slightly larger error estimates on the validation set, as the PC reconstruction



process is unaware of this data set. However, because almost all error estimates fall below 10% range, and in light of the close agreement (about 4% difference) between the blue and red dots, our PC surrogate models are deemed to suitably reproduce source model PGV responses throughout the entire station network.

Apart from the above error estimates, the convergence of PC surrogate models with respect to truncation order is also investigated from a statistical point of view. Figure 4 shows PGV distributions from PC re-sampling on a one-million-member LHS set ($\mathcal{P}_{LHS}^{1E6}$) at two selected stations, with different PC truncation orders. It is seen that when the truncation order is larger than five, the difference in the PGV prediction distributions becomes relatively small, suggesting that our ninth-order PC library is sufficient for the source model under consideration.

We finally compare distributions of PC and source model predictions, see Figure 5. It is observed that our PC surrogate models are indeed capable of reproducing PGV distributions produced from source model realizations of the validation set $\mathcal{P}_{LHS}^{valid}$. Besides, the excellent agreement between the two PC predicted distribution curves in Figure 5 suggests that our existing 8000 model simulation ensemble is statistically representative, which provides additional confidence in our PC representations.

## 3.2 PC Statistics

The PC surrogate models obtained in the previous section provide immediate access to prediction statistics, as given by Equations (4) and (5). Figure 6 shows means and standard deviations of PC PGV predictions at different stations, along with a reference PGV curve provided by the GMPE in Boore and Atkinson (2008). It is seen that our PC predictions generally scatter around the GMPE curve. The PGVs are generally largest near the fault plane, and decrease with increasing $R_{JB}$ distance. The overall tendency of PC prediction uncertainty (quantified by the standard deviation bars) seems to decrease with increasing $R_{JB}$ distance as well.

The conditional mapping from random PC parameter $\boldsymbol{\xi}$ to physical fault plane configuration leads to complex dependency of PGV responses to random inputs. To identify the relative impact of each random parameter (each component of $\boldsymbol{\xi}$) on model responses, we rely on the global sensitivity analysis in (Homma and Saltelli, 1996; Sobol, 1993).

Figure 7 shows both the first and total order sensitivity indices associated with each random parameter at different stations. These sensitivity indices reveal that the model PGV response is most sensitive to the location of the hypocenter ($x_h$ is dominant and $z_h$ plays a secondary role) throughout all stations, whereas the remaining random parameters (associated with elliptical asperity patch) are relatively insignificant. While it might be reasonable to neglect the elliptical patch parameters' impact on PGV response variability at far stations (with $R_{JB}$ distance roughly more than 10 km away from the center), it is evident that at near-the-center stations, those elliptical patch parameters can still lead to a considerable impact on PGV response.

To better illustrate the above sensitivity observation, we divided the parameters into the following two groups $\boldsymbol{\xi}_{hypo} = \{\xi_2^{x_h}, \xi_3^{z_h}\}$ and $\boldsymbol{\xi}_{ellip} = \{\xi_1^{AR}, \xi_4^a, \xi_5^\theta, \xi_6^{x_c}, \xi_7^{z_c}\}$ (the superscripts denote the corresponding physical parameters), and calculate the first order sensitivity indices associated with $\boldsymbol{\xi}_{hypo}$ and $\boldsymbol{\xi}_{ellip}$ using Equation (7a), denoted as $\mathbb{S}_{hypo}$ and $\mathbb{S}_{ellip}$, respectively. Note the combined effect (interaction) of hypocenter location and elliptical patch parameters is simply given by $\mathbb{S}_{hypo \times ellip} = 1 - \mathbb{S}_{hypo} - \mathbb{S}_{ellip}$. The resulting group sensitivity indices are shown in Figure 8. It is now clear that the hypocenter location alone is responsible for 80-90% of the variability in PGVs at distant stations. Meanwhile near the center, the hypocenter location



**Figure 4.** PC predicted PGV distributions at two selected stations. (Top) Station # 3 (Bottom) Station # 21. Distribution curves are generated from PC realizations on a one-million-member LHS set $\mathcal{P}_{LHS}^{1E6}$.







**Figure 5.** Comparison of PGV distributions predicted by the source model (blue solid curve) and PC surrogate model (red dashed curve) respectively at selected stations over the validation sample set $\mathcal{P}_{LHS}^{valid}$. The black dash-dotted curves are PC prediction distributions obtained from realizations on a one-million-member LHS set $\mathcal{P}_{LHS}^{1E6}$.



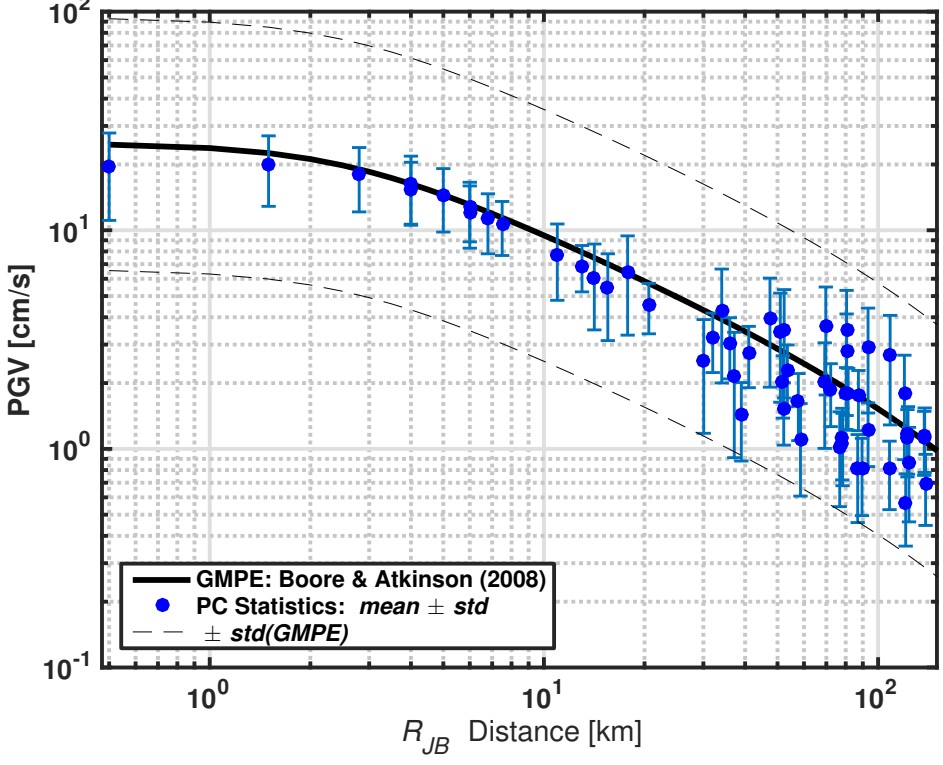

**Figure 6.** Comparison of PC statistics (based on uniform distribution assumption of PC random parameter) with GMPE results. Dashed lines are standard deviation bounds of GMPE predictions.

alone is associated with only 55-75% of the PGV variability, suggesting that the elliptical patch parameters play important roles with about 25-45% contribution to the total PGV variability.

## 4 Bayesian Inference

In this section, we utilize a Bayesian approach (Bernardo and Smith, 2001; Berger, 2013; Gelman et al., 2014) to find the most likely fault plane configuration, in the sense that the resulting earthquake rupture model produces PGVs best match the reference GMPE curve by Boore and Atkinson (2008) for the same magnitude and focal mechanism. To this end, we first obtain the GMPE predicted PGVs at the stations shown in Figure 2, denoted as $\boldsymbol{d}$, which serves as observational data in our Bayesian inference, and compare $\boldsymbol{d}$ with our PC surrogate model predictions $\tilde{\boldsymbol{d}}(\boldsymbol{\xi}) = (\tilde{\mathcal{Q}}_1(\boldsymbol{\xi}), \tilde{\mathcal{Q}}_2(\boldsymbol{\xi}), ..., \tilde{\mathcal{Q}}_{N_{obs}}(\boldsymbol{\xi}))^T$.





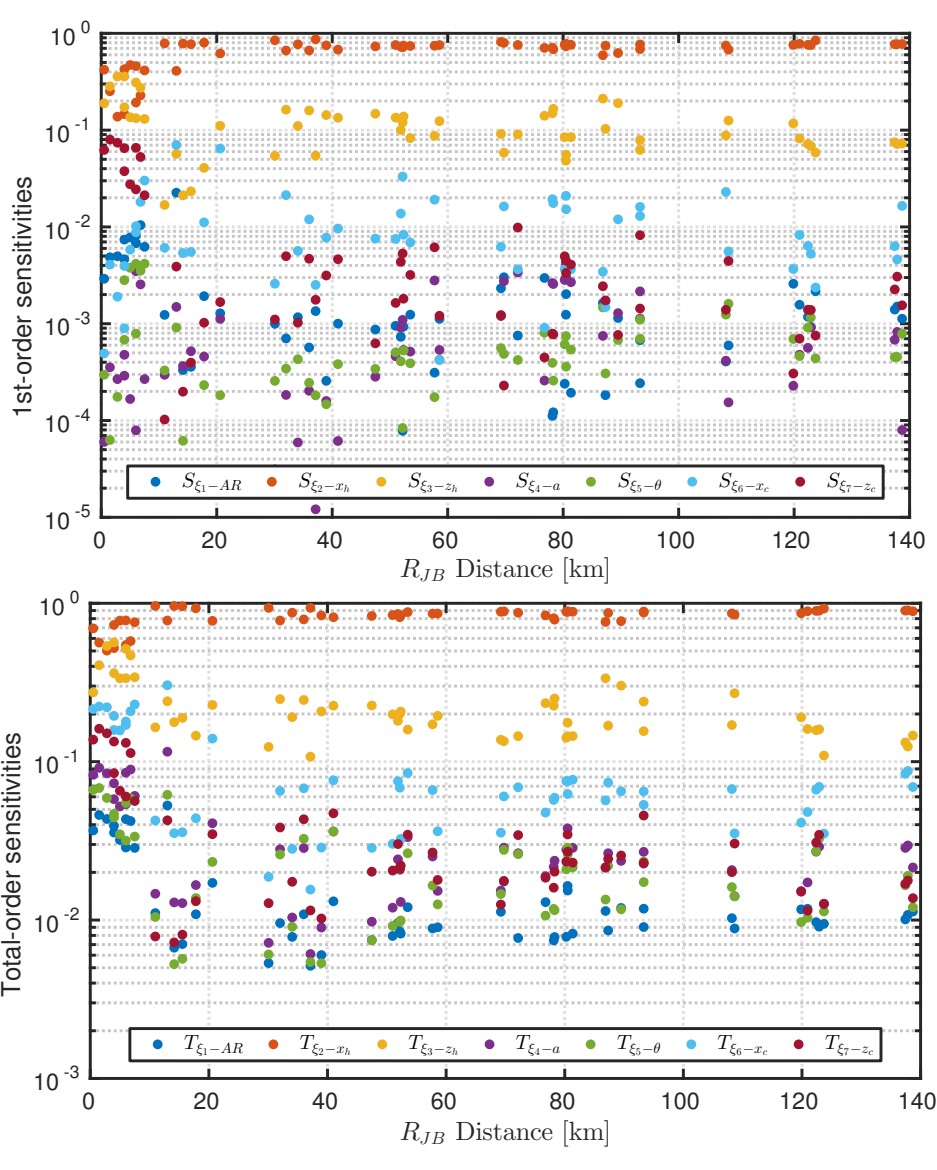

**Figure 7.** First (top) and total (bottom) order sensitivity indices at each station.



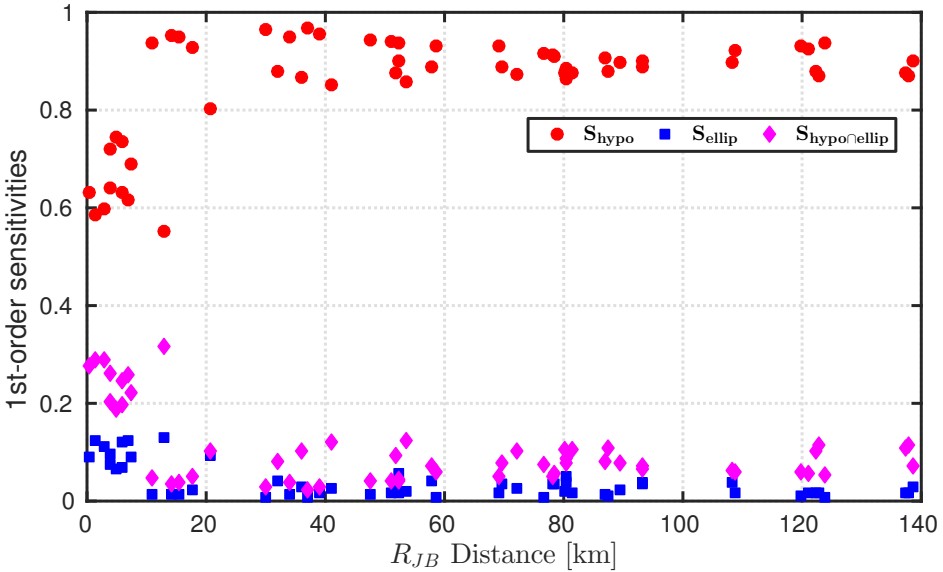

**Figure 8.** 1st order sensitivity indices with respect to grouped parameters.

## 4.1 Bayesian Formulation

To formulate the Bayesian problem, we start with Bayes' formula

$$p(\boldsymbol{\eta}|\boldsymbol{d}) = \frac{p(\boldsymbol{d}|\boldsymbol{\eta})p(\boldsymbol{\eta})}{p(\boldsymbol{d})} \propto p(\boldsymbol{d}|\boldsymbol{\eta})p(\boldsymbol{\eta}), \tag{9}$$

where $\boldsymbol{\eta}$ is the parameter vector to be inferred, $p(\boldsymbol{\eta})$ is the prior probability distribution of $\boldsymbol{\eta}$, and $p(\boldsymbol{d}|\boldsymbol{\eta})$ is the likelihood of observing $\boldsymbol{d}$ given $\boldsymbol{\eta}$. The denominator $p(\boldsymbol{d})$ is the marginal distribution known as evidence. (Note this evidence can be neglected, as the Markov Chain Monte Carlo (MCMC) sampling method (Haario et al., 2001; Roberts and Rosenthal, 2009) utilized below solely relies on the proportionality). We adopt the assumption of independent Gaussian prediction error at each station location, i.e. the discrepancy between GMPE and PC predictions at each station is an independent Gaussian variable:

$$p(\epsilon_j) = p(d_j - \tilde{d}_j) = \frac{1}{\sqrt{2\pi\sigma^2}}exp\Big[-\frac{(d_j - \tilde{d}_j)^2}{2\sigma^2}\Big], \quad \forall j \in \{1, 2, ..., N_{obs}\}. \tag{10}$$

Recall that the PC prediction uncertainty seems to decrease with $R_{JB}$ distance according to Figure 6, which motivates us to partition the $N_{obs}$ stations into four concentric groups, namely according to their corresponding $R_{JB}$ distances as colored in Figure 2, and associate each group of stations with a hyper-parameter $\sigma^2_{l(j)}$ ($l(j) \in \{1, 2, 3, 4\}$, depending on the $R_{JB}$ distance of the $j$-th station). As a result, the likelihood can be expressed as:

$$p(\boldsymbol{d}|\boldsymbol{\eta}) = \prod_{j=1}^{N_{obs}} \frac{1}{\sqrt{2\pi\sigma^2_{l(j)}}}exp\Big(-\frac{(d_j - \tilde{d}_j(\boldsymbol{\xi}))^2}{2\sigma^2_{l(j)}}\Big), \tag{11}$$



and accordingly the inference parameter vector $\boldsymbol{\eta}$ reads

$$\boldsymbol{\eta} = (\xi_1, \xi_2, ..., \xi_7, \sigma_1^2, \sigma_2^2, ..., \sigma_4^2)^T. \tag{12}$$

Our numerical experiments suggest that the 4-$\sigma^2$ model above outperforms the model with only one hyper-parameter for all stations. It is noted that we limit the number of uncertainty hyper-parameters ($\sigma_i^2$'s) to four in this study, due to the limited number of observations (PGVs at limited number of stations). If more observations are available, it might be beneficial to increase the number of hyper-parameters.

The prior distribution of $\boldsymbol{\eta}$, without additional information on the model parameters, is usually given by assumptions of uniform distribution for PC parameters $\boldsymbol{\xi}$, and Jeffrey's priors (Sivia and Skilling, 2006) for hyper-parameters $\sigma_l^2$ (as $\sigma_l^2$ is always greater than zero); consequently,

$$p(\boldsymbol{\eta}) = \begin{cases} \left(\frac{1}{2}\right)^7 \prod_{l=1}^4 \frac{1}{\sigma_l^2} & \forall \boldsymbol{\xi} \in \boldsymbol{\Xi} \text{ and } \forall \sigma_l^2 > 0, \\ 0 & \text{otherwise,} \end{cases} \tag{13}$$

and Bayes' rule reduces to

$$p(\boldsymbol{\eta}|\boldsymbol{d}) \propto p(\boldsymbol{d}|\boldsymbol{\eta})p(\boldsymbol{\eta}) =$$

$$\begin{cases} \prod_{j=1}^{N_{obs}} \frac{1}{\sqrt{2\pi\sigma_{l(j)}^2}} exp\left(-\frac{(d_j - \tilde{d}_j(\boldsymbol{\xi}))^2}{2\sigma_{l(j)}^2}\right) \left[\left(\frac{1}{2}\right)^7 \prod_{l=1}^4 \frac{1}{\sigma_l^2}\right] & \forall \boldsymbol{\xi} \in \boldsymbol{\Xi} \text{ and } \forall \sigma_l^2 > 0, \\ 0 & \text{otherwise.} \end{cases} \tag{14}$$

We rely on the adaptive metropolis MCMC approach (Haario et al., 2001; Roberts and Rosenthal, 2009) to sample the above posterior distribution. It is worth noting that MCMC methods, despite the improved efficiency against the traditional MC approaches, generally require a large number of samples (typically tens of thousands, and even larger depending on the dimensionality of the problem). This is one of the main reasons why we utilize PC techniques, as the use of the corresponding surrogates in the MCMC simulation leads to significant reduction in computational cost. In this study, the MCMC sample size for inference is set to $10^6$.

### 4.2 Inference Results

As mentioned above, we exploit the PC surrogate models in Bayesian inference analysis and update the posterior distribution of random parameters ($\boldsymbol{\xi} \in \boldsymbol{\Xi}$), as well as PGV prediction uncertainties ($\sigma_l^2$'s), in light of the GMPE predicted PGVs. Figure 9 shows the posterior probability distributions of hyper-parameters $\sigma_l^2$ ($l \in \{1, 2, 3, 4\}$). It is evident that $\sigma_l^2$ decreases with $R_{JB}$ distance (from $l = 1$ to $l = 4$), which supports our previous ansatz from Figure 6.

Similarly, we examine the sampling chains of PC random parameters $\xi_i$ ($i \in \{1, 2, ..., 7\}$). While some parameters (e.g. $\xi_1, \xi_2, \xi_3$ and $\xi_6$) yield very informative posterior distributions (not shown here), others look relatively less informative. It is noted that our goal is to estimate the posterior distributions of the physical parameters in Table 2, instead of the PC parameters. Thus, it is desired to map the $\boldsymbol{\xi}$ chain into the corresponding physical configuration chain, before inferring the most likely fault plane configuration.





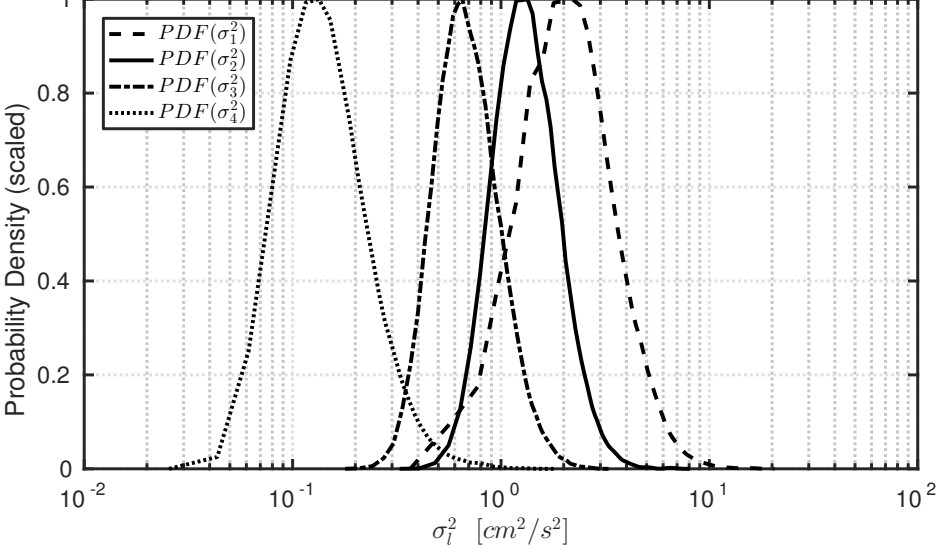

**Figure 9.** Posterior probability distributions of prediction uncertainty parameters (each PDF curve is scaled to have unit peak height for better comparison).

Figure 10 shows the posterior distributions of the physical parameters after mapping from the PC parameter chain of $\boldsymbol{\xi}$ (for brevity, the chain plots of physical parameters are not shown here), as well as the corresponding inference of the fault plane configuration (bottom right panel). It is observed that in light of the GMPE PGV predictions: 1) the hypocenter location ($x_h$

and $z_h$) is well identified; 2) The size of the elliptical patch seems to be more likely near the lower bound of the prior; 3) The inclination angle of the elliptical patch, as well as the location of the patch, is less conclusive. For example, despite the clear peak in the inclination angle plot, the posterior distribution is relatively flat, suggesting limited information gain comparing with the prior knowledge. Furthermore, the $x_c$ distribution only shows the fact that the ellipse tends to be in the left half of the fault plane; the definite location of the elliptical patch (either $x_c$ or $y_c$) is ambiguous. These findings are generally consistent

with the results of the sensitivity analysis. Since the model is primarily sensitive to the hypocenter location, perturbing the hypocenter location leads to more effective adjustment in PGV responses. On the other hand, elliptical patch parameters have relatively small impact on PGV variance, which calls for more observational data to pin down those parameters.

One needs to be cautious about the Bayesian inference results discussed above. From the physical point of view, the spatial distribution of those stations (see Figure 2) where PGVs are reported is almost 'symmetric' about the center of the fault plane

($x = 0$ and $y = 0$), as a result, one would expect to see a 'symmetric' twin configuration that are roughly equally plausible from the Bayesian inference. However, this 'symmetric' counterpart is clearly missing in the above inference results. This is probably because when MCMC chain converges to the high probability region of hypocenter location in the bottom right quadrant of the fault plane, it becomes more and more difficult to escape from this high probability region and explore the other side of parameter space. In other words, there could be bi-modal structures in the distributions of $x_h$ (as well as $x_c$)





**Figure 10.** Prior (dashed black, derived from uniform $\boldsymbol{\xi}$ distribution in $\Xi$) and posterior (solid blue) distributions of physical fault plane configuration parameters. The bottom right panel shows the inferred fault plane configuration.





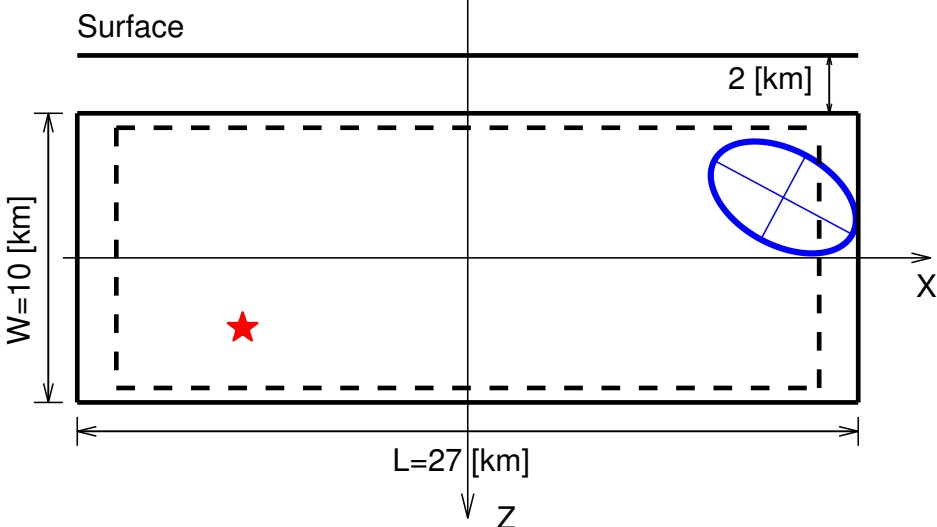

**Figure 11.** Inferred fault plane configuration with MCMC chain starting from the 'symmetric' counterpart configuration.

which the previous MCMC process fails to identify (e.g. the configuration in which the hypocenter located on the bottom left quadrant of the fault plane, and the ellipse centered at somewhere in the right half of the fault plane). While in theory it is possible to identify the missing multi-modal distributions of random parameters by further increasing the number of MCMC samples, the computational cost can be excessive. Alternatively, we verify our expectation of seeing the 'symmetric' counterpart configuration by re-running the MCMC simulation starting with the 'symmetric' counterpart configuration (i.e. with hypocenter being in the bottom left quadrant of the fault plane, and elliptical patch being in the right side of the fault plane). The resulting fault plane configuration inference is shown in Figure 11. As expected, the new MCMC process ended up with a fault plane configuration that is roughly 'symmetric' to the previous inference result, especially for the hypocenter location. The asymmetric behavior of the elliptical patch stems from the fact that: 1) the $N_{obs}$ stations are not exactly symmetrically distributed, thus one should not expect exact symmetry; 2) as discussed before, the PGV responses are less sensitive to the elliptical patch properties, leading to ambiguity in inferring these properties.

### 4.3 Inference with Restricted Prior

The previous inference results are all based on almost complete ignorance of dependency between hypocenter location and the slip area (asperity). However, previous studies (Mai et al., 2005; Irikura and Miyake, 2011) suggested some constraints on the relative hypocenter location (Mai et al., 2005) with respect to the asperity, and size of the asperity (Irikura and Miyake, 2011). In this section, we consider the following restrictions in our inference analysis:

R-1. The elliptical patch is inside the dashed rectangle ($[L', W'] = 0.9 \times [L, W]$) shown in Figure 1;

R-2. The area ratio of the elliptical patch ($AR$) is between 15% and 29% of the fault plane area, i.e. $0.15 < AR < 0.29$;





R-3. The elliptical patch is not too elongated, i.e. the axis ratio $\frac{a}{b} \leq 3$;

R-4. The hypocenter is located outside but near the elliptical patch, i.e. $x_h = (a+3\zeta_{h_1})cos(2\pi\zeta_{h_2})$ and $z_h = (b+b\frac{3}{a}\zeta_{h_1})sin(2\pi\zeta_{h_2})$ $\forall(\zeta_{h_1}, \zeta_{h_2}) \in [0,1]^2$.

The above restrictions can be conveniently incorporated into the Bayesian framework, namely by modifying the previous prior distribution (Equation (13)) as follows:

$$p^*(\boldsymbol{\eta}) = \begin{cases} \left(\frac{1}{2}\right)^7 \prod_{l=1}^4 \frac{1}{\sigma_l^2} & \forall\boldsymbol{\xi} \in \boldsymbol{\Xi} \ , \ \forall\sigma_l^2 > 0 \ \text{ and all restrictions are satisfied,} \\ 0 & \text{otherwise.} \end{cases} \tag{15}$$

However, due to the strong restrictions listed above, the support of the above prior probability distribution (Equation (15)) turns out to be extremely limited in the parameter space $\boldsymbol{\Xi}$, leading to computationally inefficient MCMC sampling (since most of the samples drawn from a proposal distribution will end up not satisfying at least one of the restrictions and thus zero prior probability). To mitigate the difficulty of inefficient sampling due to restricted prior distribution, we introduce a new layer of parameterization, mapping from $\boldsymbol{\Xi}$ to restricted physical configurations. (Details on this new mapping mechanism are given in appendix B.)

Figure 12 shows the MCMC process of drawing random samples from proposal distributions and calculate the resulting posterior probability. Without additional restrictions (orange path), the parameter vector $\boldsymbol{\zeta} = \boldsymbol{\xi}$, and the whole process reduces to the standard MCMC process we used in the previous section. By introducing the new parameterization process (see algorithm 2), we are transforming the original problem, which is based on PC parameter vector $\boldsymbol{\xi}$, into a new inference problem based on $\boldsymbol{\zeta}$ (we denote $\boldsymbol{\zeta}$ as auxiliary random parameter vector hereafter, to distinguish it from the PC parameter vector $\boldsymbol{\xi}$). This transformation is based on the mapping from $\boldsymbol{\zeta}$ to $\boldsymbol{\xi}$ (i.e. $\boldsymbol{\xi} = \boldsymbol{\xi}(\boldsymbol{\zeta})$) via their commonly associated physical configuration. For clarity, we formulate the new $\boldsymbol{\zeta}$ based Bayesian problem as follows:

$$p(\boldsymbol{\eta}^*|\boldsymbol{d}) \propto \begin{cases} \left[\left(\frac{1}{2}\right)^7 \prod_{l=1}^4 \frac{1}{\sigma_l^2}\right] \prod_{j=1}^{N_{obs}} \frac{1}{\sqrt{2\pi\sigma_{l(j)}^2}} exp\left(-\frac{(d_j - \tilde{d}_j(\boldsymbol{\xi}(\boldsymbol{\zeta})))^2}{2\sigma_{l(j)}^2}\right) & \forall\boldsymbol{\zeta} \in \boldsymbol{\Xi}, \forall\sigma_l^2 > 0, \\ 0 & \text{otherwise.} \end{cases} \tag{16}$$

where $\boldsymbol{\eta}^* = (\zeta_1, \zeta_2, ..., \zeta_7, \sigma_1^2, \sigma_2^2, ..., \sigma_4^2)^T$.

Following the same analysis as discussed before, we show the inference results under restrictions in Figure 13. Note that the prior distributions of those physical parameters are different from those in Figure 10, as the new ones are derived from uniformly distributed auxiliary random vector $\boldsymbol{\zeta} \in \boldsymbol{\Xi}$, instead of PC parameters $\boldsymbol{\xi} \in \boldsymbol{\Xi}$. Nevertheless, we see very consistent results of hypocenter location, as well as the location of the elliptical patch, comparing with those in Figure 10. The area aspect ratio $AR$, though larger than the previous inferred value, still favors the lower end of the prescribed parameter range. The elliptical patch ends up with a larger area and longer semi-major axis (compared to the results in Figure 10 and 11). These differences are directly stemming from restrictions R-2 and R-3.

Though it is not obvious to see from Figure 13, the restricted Bayesian MCMC process is indeed aware of the existence of the 'symmetric' counterpart configuration. Figure 14 shows the restricted Bayesian MCMC sample chains of both the hypocenter





**Figure 12.** Flow chart demonstrating the random sampling process and the calculation of posterior probability in MCMC. The orange path corresponds to unrestricted sampling process, whereas the blue path incorporates additional restrictions on fault plane configurations. Note $Y$ denotes the fault plane configuration vector in the physical domain, e.g. $Y = (AR, x_h, z_h, a, \theta, x_c, z_c)^T$.







**Figure 13.** Prior (dashed black, derived from uniform $\zeta$ distribution in $\Xi$) and posterior (solid blue) distributions of physical fault plane configuration parameters in restricted inference. The bottom right panel shows the inferred fault plane configuration.





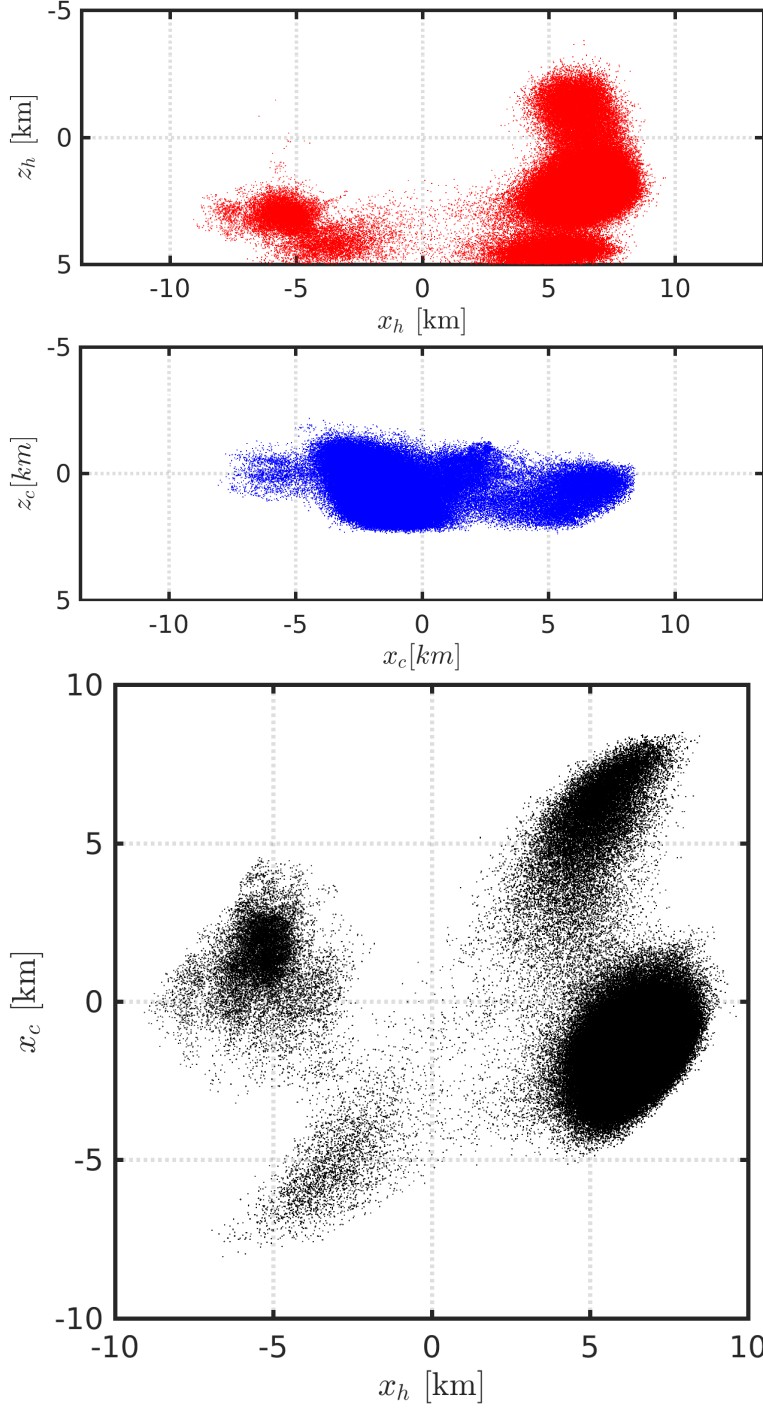

**Figure 14.** Restricted Bayesian MCMC sample chains of the hypocenter (top) and elliptical patch center (middle); the bottom panel shows the correspondence between $x_h$ and $x_c$ chains





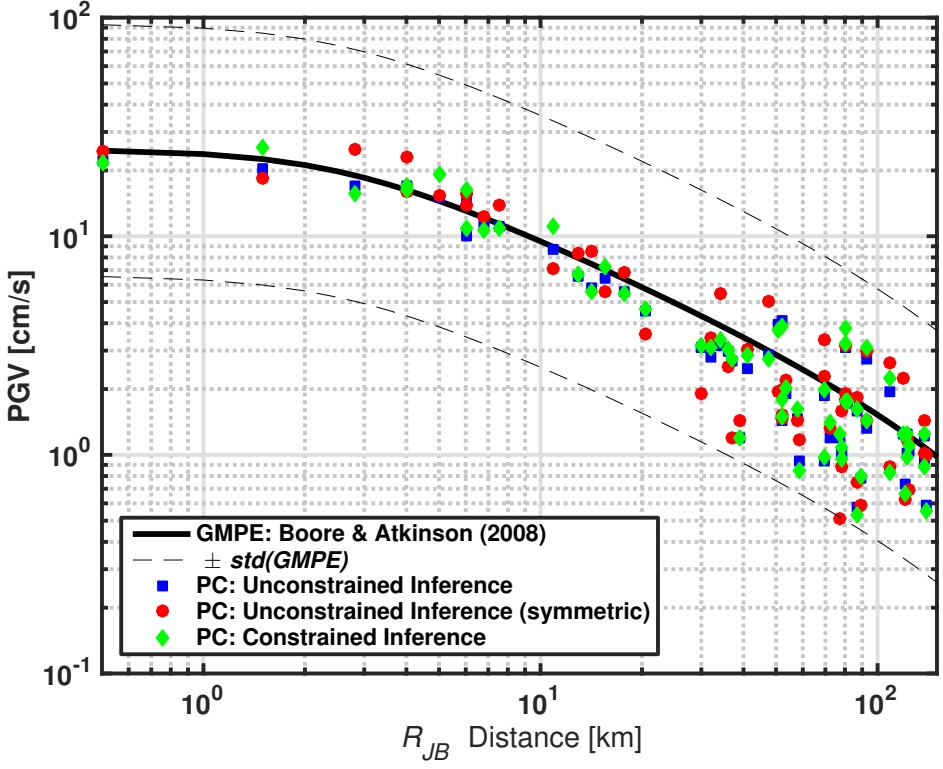

**Figure 15.** Comparison of PC predicted PGV responses with aforementioned three inferred fault plane configurations with the reference GMPE curve. Dashed lines are standard deviation bounds of GMPE predictions.

(top panel) and elliptical patch center (middle panel). It is seen that despite the fact the hypocenter samples are mostly clustered around $x_h = 5$ km, there is a sample cloud on the opposite side ($x_h = -5$ km), corresponding to the 'symmetric' counterpart

25  configuration discussed before. The sample cloud of elliptical center also shows bi-modal distributions, with primary cloud on the left ($x_c < 0$) and secondary 'symmetric' counterpart on the right (around $x_c = 5$ km). The correspondence between $x_h$ and $x_c$ is shown in the bottom panel of Figure 14, from which it is seen that when $x_h$ is positive, $x_c$ is more likely to be negative and vice versa, suggesting that hypocenter and ellipse center are in the opposite side of the fault plane, as previous inference results suggested. Note that in this restricted Bayesian MCMC sampling, the total number of samples remains $10^6$. The ability

30  to observe the 'symmetric' counterpart clouds is probably due to the fact that by introducing the auxiliary parameter $\zeta$, we dramatically shrunk the sampling space (it is only a small subspace of the original unrestricted parameter space). As mentioned before, introducing the auxiliary parameter $\zeta$ leads to significant efficiency improvement in MCMC sampling process.

### 4.4 Comparing PGVs

We summarize the Bayesian analysis by comparing PC predicted PGV responses to the three inferred fault plane configurations discussed above with the reference GMPE curve (see Figure 15 and Table 3). We observe that all three configurations lead to





**Table 3.** Comparison of PC predicted PGVs of different inferred configurations with the reference GMPE curve. Unrestricted-1 and 2 correspond to inferences in Figure 10 and Figure 11, respectively.

| Inference | $\epsilon = \sqrt{\frac{\sum_{j=1}^{N_{obs}} (\tilde{\mathcal{Q}}_j - \mathcal{Q}_j^{GMPE})^2}{N_{obs}}}$ | $r = \sqrt{\frac{1}{N_{obs}} \sum_{j=1}^{N_{obs}} \left( \frac{\tilde{\mathcal{Q}}_j - \mathcal{Q}_j^{GMPE}}{\mathcal{Q}_j^{GMPE}} \right)^2}$ |
|---|---|---|
| Unrestricted-1 (blue) | 1.1135 | 0.3395 |
| Unrestricted-2 (red) | 1.7413 | 0.3993 |
| Restrict (green) | 1.4564 | 0.3702 |

relatively close match between PGV responses and the reference GMPE curve. By comparing either the root-mean-square (rms) error or the relative rms error, we conclude that the red dots (corresponding to the unrestricted inference in Figure 11) clearly show larger discrepancy from the GMPE curve, suggesting smaller likelihood compared to the other two, consistent with our Bayesian analysis. When comparing the blue and green dots (unrestricted inference in Figure 10 versus restricted inference in Figure 13), the former seems to be slightly better, which is expected because of the additional flexibility in fitting the GMPE curve. Nevertheless, it might be better to report the restricted inference results (configuration in Figure 13), as it satisfies all the restrictions learned from previous studies while retaining plausible agreement with the reference GMPE curve.

## 5   Conclusions

An earthquake rupture model was adopted to explore the stochastic dependence of ground motions (in terms of PGVs) on random fault plane configurations. Thanks to the ability to generate two independent source model simulation ensembles with 8000 members each, we were able to build successful PC surrogate models to assess PGV responses over the virtual network of $N_{obs} = 56$ stations from one ensemble, and then to validate the quality of PC models on the other. Our statistical analysis showed that the two 8000-member LHS ensembles of source model simulations are adequate to represent the underlying PGV distributions at all stations, as they closely match with PC predicted distributions over a much larger sample set.

A global sensitivity analysis of PC surrogate models was conducted. The analysis revealed that the source model PGV response is primarily sensitive to the hypocenter location, and much less sensitive to properties of the asperity patch, especially at stations far away from the fault plane (in terms of the $R_{JB}$ distance). While this holds true for all stations, it is noted that asperity patch properties still carry considerable impact (20-30% associated variability) on PGV responses at stations close to the fault plane, and even more influence (additional 10% variability) if one takes into consideration the interaction between asperity patch and hypocenter location.

Our analysis of PGV variabilities indicated that one needs to be cautious when interpreting PGVs at near fault plane stations, as they are more prone to higher model noise. This is supported by the Bayesian inference analysis, in which four independent model noise parameters ($\sigma_l^2$ for $l = 1, 2, 3, 4$) were introduced and assigned to four concentric groups of observational stations, depending on their $R_{JB}$ distances away from the fault plane. The Bayesian inference results clearly showed the decreasing trend of noise parameters ($\sigma_l^2$'s) when moving away from the fault plane (see Figure 9). Further refinement of the noise





parameter profile along the $R_{JB}$ distance, though desired, is prohibited by the limited number of available observational

30  stations.

We conducted both unrestricted and restricted Bayesian inference analyses to identify the chosen GMPE reference curve. The key findings are as follows: 1) due to the considerable 'symmetry' presented by those $N_{obs}$ stations, the most profound fault plane configuration, which reproduce the reference GMPE predictions, can potentially have a 'symmetric' twin configuration, especially for the hypocenter location; 2) it is more likely to have the hypocenter located in the lower right quadrant of the fault plane, and the elliptical patch centered in the lower left quadrant; 3) the restricted inference results remain consistent with the unrestricted ones, with slightly more deviation from the chosen GMPE reference curve but closer agreement with the previous study (Mai et al., 2005).

The analyses and findings in this study provide useful insights on how near-source ground shaking (and its variability) depend on random fault rupture configurations. Interestingly, even very simple source models (with elliptical slip patches) are able to generate shaking distributions that well reproduce empirical predictions. To better reproduce the chosen GMPE reference curve, it might be beneficial to consider two or more asperity patches, instead of one in this study, in order to reduce the hypocenter location influence and in return increase the impact of asperity properties. Another potential improvement can

be made by refining the station network. As mentioned earlier, the Bayesian inference is primarily limited by the number of available stations at which PGVs are reported. By increasing the number of PGV reporting stations, one may improve the Bayesian inference results (e.g. removing the ambiguity in inferring the elliptical patch location).

*Code and data availability.* The COMPSYN code (Spudich and Xu, 2003) employed in this study, along with the simulation data are available upon request.

**Appendix A: Mapping from PC Random Parameters to Physical Parameters**

Let $a$ and $b$ be the lengths of semi-major and minor axes, respectively, of the elliptical patch considered in the fault plane configuration discussed in Section 2, and $AR$ be the area aspect ratio defined by $AR = \frac{\pi ab}{LW}$ (here $L = 27km$ and $W = 10km$ are the length and width of the fault plane). The elliptical patch centered at the origin ($x_c = 0$ and $z_c = 0$, note the $z$-axis is pointing downwards as shown in Figure 1), when not rotated (meaning $\theta = 0$, the semi-major axis align with x-axis), can be

expressed as:

$$
\begin{bmatrix} x \\ z \end{bmatrix} = \begin{bmatrix} a cos\beta \\ b sin\beta \end{bmatrix} \quad \text{where} \quad -\pi \leq \beta \leq \pi \tag{A1}
$$

If the elliptical patch is rotated by $\theta \in [-30°, +30°]$ (a positive angle denotes clockwise rotation), then the ellipse is given by:

$$
\begin{bmatrix} x^r \\ z^r \end{bmatrix} = \begin{bmatrix} cos\theta & -sin\theta \\ sin\theta & cos\theta \end{bmatrix} \begin{bmatrix} x \\ z \end{bmatrix} = \begin{bmatrix} cos\theta & -sin\theta \\ sin\theta & cos\theta \end{bmatrix} \begin{bmatrix} a cos\beta \\ b sin\beta \end{bmatrix} = \begin{bmatrix} a cos\theta cos\beta - b sin\theta sin\beta \\ a sin\theta cos\beta + b cos\theta sin\beta \end{bmatrix} \tag{A2}
$$





To ensure the resulting elliptical patch is completely confined within the fault plane, we first find the maximum extent of the

ellipse in both x- and y-directions. We first calculate the following two $\beta^*$'s,

$$
\begin{aligned}
\frac{\partial x^r}{\partial \beta} = -a\cos\theta\sin\beta - b\sin\theta\cos\beta = 0 \;\; &\Rightarrow \;\; \beta_x^* = \tan^{-1}\Big(-\frac{b}{a}\tan\theta\Big) \\
\frac{\partial z^r}{\partial \beta} = -a\sin\theta\sin\beta + b\cos\theta\cos\beta = 0 \;\; &\Rightarrow \;\; \beta_z^* = \tan^{-1}\Big(\frac{b}{a}\frac{1}{\tan\theta}\Big)
\end{aligned}
\tag{A3}
$$

Next, by substitute the above $\beta_x^*$ and $\beta_z^*$ into Equation (A2), we have

$$
\begin{aligned}
x_{max}^r &= |a\cos\theta\cos\beta_x^* - b\sin\theta\sin\beta_x^*| \\
z_{max}^r &= |a\sin\theta\cos\beta_z^* + b\cos\theta\sin\beta_z^*|
\end{aligned}
\tag{A4}
$$

These are the maximum extents of the ellipse in x- and y- directions, respectively.

When the ellipse is not centered at the origin ($x_c \neq 0$ and/or $z_c \neq 0$), the following conditions need to be satisfied.

$$
\begin{aligned}
|x_c| + x_{max}^r &\leq \frac{L}{2} \\
|z_c| + z_{max}^r &\leq \frac{W}{2}
\end{aligned}
\tag{A5}
$$

which leads to:

$$
\begin{aligned}
|x_c| &\in [0, \frac{L}{2} - x_{max}^r] \\
|z_c| &\in [0, \frac{W}{2} - z_{max}^r]
\end{aligned}
\tag{A6}
$$

Note the above constraint on $x_c$ is always valid, since $x_{max}^r \leq a \leq \frac{L}{2}$; while the $z_c$ constraint requires more treatment as $z_{max}^r$

can be greater than $\frac{W}{2}$ under some rotation angle $\theta$ and semi-major axis $a$. To ensure that $z_{max}^r \leq \frac{W}{2}$, we first check if the

prescribed upper bound rotation ($30°$) is feasible. If not, we solve the following equation for $\theta^*$, which corresponding to the

maximum feasible rotation angle given $a$ and $AR$.

$$
z_{max}^r = |a\sin\theta^*\cos\beta_z^*(\theta^*, a, AR) + b\cos\theta^*\sin\beta_z^*(\theta^*, a, AR)| = \frac{W}{2}
\tag{A7}
$$

and define the upper bound of the rotation angle as

$\hat{\theta} = min(\theta^*(PE, a), 30°)$  (A8)

The resulting rotation angle parameter $\theta$ is then assumed to be uniformly distributed over $[-\hat{\theta}, \hat{\theta}]$.

The mapping from $\boldsymbol{\xi}$ to physical parameters is outlined in the Algorithm 1. With the prior assumption of uniform distribution

of $\boldsymbol{\xi}$ in $\Xi$, the corresponding prior distributions of each physical parameter are show in Figure 10 (dashed black curves).



---

**Algorithm 1** Unrestricted mapping - PC random parameter $\boldsymbol{\xi}$ to physical parameters: $\boldsymbol{Y} = \mathcal{M}_1(\boldsymbol{\xi})$

---

1: Input $\forall \boldsymbol{\xi} = (\xi_1, \xi_2, ..., \xi_7)^T \in \boldsymbol{\Xi}$

2: $AR = 0.05 + \frac{1}{2}(\xi_1 + 1)(0.29 - 0.05)$      {Map $\xi_1$ to area ratio}

3: $x_h = -\frac{L}{2} + \frac{1}{2}(\xi_2 + 1)L$      {Map $(\xi_2, \xi_3)$ to hypocenter location $(x_h, z_h)$}

4: $z_h = -\frac{W}{2} + \frac{1}{2}(\xi_3 + 1)W$

5: $a_{min} = \sqrt{\frac{AR \cdot L \cdot W}{\pi}}$      {Calculate the lower bound of $a$ from $AR$ above}

6: $a = a_{min} + \frac{1}{2}(\xi_4 + 1)(\frac{L}{2} - a_{min})$      {Map $\xi_4$ to $a$, and calculate $b$}

7: $b = \frac{AR \cdot L \cdot W}{\pi a}$

8: **if** $z_{max}^r(a, b, 30°) > \frac{W}{2}$ **then**

9:      Solve Equation (A7) for $\theta^*$

10:      let $\hat{\theta} = \theta^*$      {Calculate maximum feasible rotation angle $\hat{\theta}$}

11: **else**

12:      let $\hat{\theta} = 30°$      {Prescribe maximum feasible rotation angle otherwise}

13: **end if**

14: $\theta = -\hat{\theta} + \hat{\theta}(\xi_5 + 1)$      {Map $\xi_5$ to rotation $\theta$}

15: Plug $(a, b, \theta)$ into Equation (A4) to calculate $x_{max}^r$ and $z_{max}^r$

16: $x_c \in [x_c^{min}, x_c^{max}] = [-\frac{L}{2} + x_{max}^r, \frac{L}{2} - x_{max}^r]$

17: $z_c \in [z_c^{min}, z_c^{max}] = [-\frac{W}{2} + z_{max}^r, \frac{W}{2} - z_{max}^r]$

18: $x_c = x_c^{min} + \frac{1}{2}(\xi_6 + 1)(x_c^{max} - x_c^{min})$      {Map $(\xi_6, \xi_7)$ to ellipse center $(x_c, z_c)$}

19: $z_c = z_c^{min} + \frac{1}{2}(\xi_7 + 1)(z_c^{max} - z_c^{min})$

20: return $\boldsymbol{Y} = (AR, x_h, z_h, a, \theta, x_c, y_c)^T$      {Return parameter vector in the physical domain}

---

## Appendix B: Restricted Mapping

We introduce the auxiliary parameter vector $\boldsymbol{\zeta} \in \boldsymbol{\Xi}$, and design the following mapping process to generate fault plane configuration samples that satisfy our prior configuration restrictions. For clarity, we list again the four restrictions below:

5    R-1.  The elliptical patch is inside the dashed rectangle ($[L', W'] = 0.9 \times [L, W]$) shown in Fig. 1;

R-2.  The area of the elliptical patch ($AR$) is between 15% and 29% of the fault plane area, i.e. $0.15 < AR < 0.29$;

R-3.  The elliptical patch is not too elongated, i.e. $\frac{a}{b} < 3$;

R-4.  The hypocenter is located outside but near the elliptical patch, i.e. $x_h = (a + 3\zeta_{h_1})cos(2\pi\zeta_{h_2})$ and $z_h = (b + b\frac{3}{a}\zeta_{h_1})sin(2\pi\zeta_{h_2})$   $\forall(\zeta_{h_1}, \zeta_{h_2}) \in [0, 1]^2$;

    The mapping process is similar to the one in Algorithm 1, with necessary modifications to satisfy the above conditions. We outline the constrained mapping in Algorithm 2. Note there is one additional condition needs to be verified, i.e. whether or not the hypocenter is inside the fault plane, as it is not guaranteed by the mapping process (this is also indicated in Figure 12).





---

**Algorithm 2** Restricted mapping - auxiliary parameter vector $\boldsymbol{\zeta}$ to physical parameters: $\boldsymbol{Y} = \mathcal{M}_2(\boldsymbol{\zeta})$

---

1: Input $\forall \boldsymbol{\zeta} = (\zeta_1, \zeta_2, ..., \zeta_7)^T \in \Xi$

2: $[L', W'] = 0.9 \times [L, W]$          {Set the restricted rectangle dimension}

3: $[AR_l^*, AR_u^*] = [\frac{0.15}{0.81}, 0.29]$     {Calculate area ratio range w.r.t $[L', W']$, the upper bound (0.29) corresponds to the   }
               maximum circle in $[L', W']$

4: $AR^* = AR_l + \frac{1}{2}(\zeta_1 + 1)(AR_u^* - AR_l^*)$          {Map $\zeta_1$ to temporary area ratio $AR^*$}

5: $a_{min} = \sqrt{\frac{AR^* \cdot L' \cdot W'}{\pi}}$          {Calculate the lower bound of $a$ from $AR^*$}

6: $a = a_{min} + \frac{1}{2}(\zeta_4 + 1)(\frac{L'}{2} - a_{min})$          {Map $\zeta_4$ to $a$, and calculate $b$}

7: $b = \frac{AR^* \cdot L' \cdot W'}{\pi a}$

8: $AR = \frac{\pi ab}{L \cdot W}$          {Calculate area ratio w.r.t the original rectangle $[L, W]$}

9: $x_h = (a + 3\frac{\zeta_2+1}{2})\cos(2\pi\frac{\zeta_3+1}{2})$

10: $z_h = (b + b\frac{3}{a}\frac{\zeta_2+1}{2})\sin(2\pi\frac{\zeta_2+1}{2})$     {Map $(\zeta_2, \zeta_3)$ to hypocenter location $(x_h, z_h)$, note the resulting $(x_h, z_h)$ can   }
               be outside the fault plane, in which case the posterior probability is set to zero.

11: **if** $z_{max}^r(a, b, 30°) > \frac{W'}{2}$ **then**

12:      Solve Equation (A7) for $\theta^*$ (using $AR^*$)          {Calculate maximum feasible rotation angle $\hat{\theta}$}

13:      let $\hat{\theta} = \theta^*$

14: **else**

15:      let $\hat{\theta} = 30°$          {Prescribe maximum feasible rotation angle otherwise}

16: **end if**

17: $\theta = -\hat{\theta} + \hat{\theta}(\zeta_5 + 1)$          {Map $\zeta_5$ to rotation $\theta$}

18: Plug $(a, b, \theta)$ into Equation (A4) to calculate $x_{max}^r$ and $z_{max}^r$

19: $x_c \in [x_c^{min}, x_c^{max}] = [-\frac{L'}{2} + x_{max}^r, \frac{L'}{2} - x_{max}^r]$

20: $z_c \in [z_c^{min}, z_c^{max}] = [-\frac{W'}{2} + z_{max}^r, \frac{W'}{2} - z_{max}^r]$

21: $x_c = x_c^{min} + \frac{1}{2}(\zeta_6 + 1)(x_c^{max} - x_c^{min})$          {Map $(\xi_6, \xi_7)$ to ellipse center $(x_c, z_c)$}

22: $z_c = z_c^{min} + \frac{1}{2}(\zeta_7 + 1)(z_c^{max} - z_c^{min})$

23: return $\boldsymbol{Y} = (AR, x_h, z_h, a, \theta, x_c, y_c)^T$          {Return parameter vector in the physical domain}

---

*Author contributions.* In this study, Hugo Cruz-Jiménez and Paul Martin Mai created the earthquake rupture model, and generated both the training and validation ensembles of model simulations for building PC surrogates. The PC based statistical analysis and Bayesian inference were conducted by Guotu Li, and Omar M. Knio. Ibrahim Hoteit provided invaluable insights and advice throughout this work.

5   *Competing interests.* The authors declare that they have no conflict of interest.

*Acknowledgements.* The authors thank the support by King Abdullah University of Science and Technology (KAUST) in Thuwal, Saudi Arabia and grants BAS/1339-01-01 for this research. The first author thanks KAUST for all support during his postdoctoral fellowship.





Earthquake rupture and ground-motion simulations have been carried out using the KAUST Supercomputing Laboratory (KSL) and we acknowledge the support of the KSL staff.



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
