# Peer review of "Bayesian inference of earthquake rupture models using polynomial chaos expansion"

_Geoscientific Model Development, 2018_

## Referee Comment (RC1) · Anonymous Referee #1 · 22 Mar 2018

in Fig. 6, the GMPE standard deviation exhibits a higher level than the PC ones. A short discussion would be interesting to explain the causes/sources of this difference.

---

## Referee Comment (RC2) · Anonymous Referee #2 · 26 Mar 2018

**General comments**

The authors develop a polynomial chaos (PC) expansion representation to provide a surrogate model for a probability distribution of Mw 6.5 strike-slip earthquakes with a fixed fault geometry. Seven parameters are used to describe a particular realization, including the hypocenter location and parameters describing an elliptical asperity, a region of relatively high slip, defining a 7-dimensional stochastic space. The surrogate model allows the rapid estimation of the peak ground velocity (PGV) at each of 56 virtual observation points. The PC expansion is computed using synthetic seismogram observations at these points for a set of 8000 realizations. A second set of 8000 realizations is used for validation, to confirm that the surrogate model constructed from the first set agrees well with the direct simulation results for the second set of realizations.

The surrogate model is then used to rapidly compute the PGV for millions of additional realizations in order to gather statistics on the decay of PGV with respect to distance from the fault (measured using the Joyner-Boore distance  $R_{JB}$ , the minimal distance to the fault plane as projected to the surface), at the 56 observation points. The mean PGV and standard deviation at each observation point are plotted vs. the distance  $R_{JB}$ , and this data compared with the ground motion prediction equation (GMPE) of Boore and Atkinson (2008). The GMPE was derived based on observations of past earthquakes and so it is interesting to see that the statistics generated by the PC expansion generally follows this prediction and lie within one standard deviation of the GMPE as determined by Boore and Atkinson. This suggests that a simplified fault model consisting of a single asperity and a small set of parameters can perhaps predict PGV statistics well, and hence may be useful for predicting other GMPE curves, or for probabilistic seismic hazard analysis more generally. The first 3 sections of the paper give a nice development of these ideas.

I had more trouble understanding the goal of Section 4, which concerns the use of Bayesian inference to determine a probability distribution on the space of PC parameters that yield an event to best match the GMPE. It seems to me that the GMPE is only intended to predict the average and standard deviation of the PGV over a large set of potential earthquakes, and so I do not understand the point of this statistical inversion to try to determine the characteristics of one particular earthquake that best matches the average. The authors conclude that the best match is more likely to have the hypocenter located in the lower right quadrant of the fault plane, and the elliptical patch centered in the lower left quadrant. Why is this useful to know? Is this meant to have geophysical significance, e.g. that real strike-slip earthquakes of this magnitude tend to have their hypocenter and asperities located in this way? How does this relate to the actual slip patterns of the real events that went into the Boore and Atkinson GMPE model, to the extent those are known? There is no discussion in the paper of these topics. I also wonder about the way this inversion is used in Section 4.5, as discussed in one of my specific comments below. I think the paper would be stronger
if the motivation for doing this inversion was better explained, since I found it hard to assess the usefulness of this part of the paper.

**Specific comments**

- 1. Page 3, line 2: The fault plane geometry is fixed with width 10 km and length 27 km. It is stated that this is obtained from 100 realizations following the scaling relation in Wells and Coppersmith (1994). How are 100 realizations used to determine these dimensions?
- 2. Page 3, lines 5–7: Why is the slip set to  $S_{max}/e$  outside the asperity? How is the slip in the asperity set? Since the area of the asperity varies with the input parameters, the slip must also vary to keep the magnitude fixed. It is stated that  $S_{max}$  varies with the ellipse size but it is not clear how.
- 3. Page 3, line 15–17: For completeness it would be good to state the grid resolution used in the COMPSYN simulation of the seismic signals, and the domain size, boundary conditions imposed, etc.
- 4. Page 3, Figure 2: The 56 observation stations surround the fault plane on all sides. Since the fault plane is vertical and the velocity model is vertically layered, shouldn't the observations be symmetric about Y = 0? If so, it would seem clearer to simply use points in the upper half plane, for example, rather than asymmetric points scattered on both sides.
- 5. Page 11, Figure 6: The points here are presumably the mean PGV observed at each of the 56 observation points, plotted vs. the distance  $R_{JB}$ . These points are calculated by evaluating the PC expansion at 1,000,000 sample points and are presumably quite accurate estimates of the mean at each observation point. But this figure shows that two points that have very similar  $R_{JB}$  can have quite
different PGV, presumably because the two points have quite different azimuthal orientation relative to the fault, even though they are the same distance away. This is interesting to observe, but since the GMPE curve ignores orientation it seems like it might also be interesting to try to average over different orientations for each distance. This could be facilitated if a number of observation points were placed at each distance, for a discrete set of distances, i.e., place the observation points on concentric rings with fixed  $R_{JB}$ . It also seems like a much larger set of observation points could be used than 56, since the PC model is so quick to evaluate. If many points were placed on many different concentric rings, then one could average over all points at a given distance to get points that might be expected to agree better with the GMPE curve in Figure 6. It would then also be possible to explore in more detail how the PGV varies with orientation along each ring.

- 6. In Figure 2 there are sets of points that have different colors/symbols that are arranged somewhat in rings, but the distance for each color do not seem to be constant. The use of colors/symbols is not explained anywhere I could find, and should be.
- 7. Page 5, Table 2: The caption says that "(\*) denotes dependent parameters". It is not clear what this means. Does this refer to the comment in line 5 of this page, where it is noted that "These restrictions lead to nonlinear dependency between feasible ranges of different physical parameters"?
- 8. The fact that some of these parameters are constrained based on the choice of other parameters means that the probability distribution of parameters is not really given by (1) on page 5 as is stated. Some choices from this 7-dimensional box have probability zero due to the constraints, while others have greater probability due to several non-allowed choices mapping to the same set of modified parameters when the asperity falls near the edge of the fault plane. Does this
affect the validity of the PC expansion and/or results? At any rate, this should be discussed.

9. Page 17, Section 4.5: In this section it is stated that a uniform distribution of parameters over the 7-dimensional space ignores various geophysical constraints suggested by previous work. This is discussed in the context of choosing a prior for the Bayesian inference, but it seems like it would be even more important to incorporate these constraints into the analysis of Section 3, where the PC expansion is used to generate statistics on the PGV for comparison with the GMPE. Why should the statistics obtained with the uniform distribution be expected to match the GMPE well if it is known that this is the wrong distribution? This is addressed to some extent in Section 4.5 where the inversion that incorporates these constraints is then used to generate statistics that are compared to the GMPE curve in Figure 15. But at this point the inversion process has been used to to further constrain the posterior distribution based on trying to match the GMPE curve, so comparing the result to the GMPE curve does not seem to provide any validation that the PC expansion could predict the GMPE curve for other scenarios, for example. I may be missing the point here, but I think it needs more explanation.

**Technical corrections**

- 1. Page 3, line 2: Presumably the rake is fixed at 0 degrees for a strike-slip event, but this should perhaps be stated?
- 2. Page 7, line 27: What are the index sets  $S_i$  and  $T_i$ ? The sets are used in the summations of (7a) and (7b) respectively, but not really defined.
- 3. Proper latex fonts for trig functions should be used in expressions such as (A1), e.g.  $a \cos \beta$  rather than  $a \cos \beta$ .

---

## Referee Comment (RC3) · Anonymous Referee #1 · 5 Apr 2018

General comments

This manuscript investigates an earthquake rupture model subject to 7 random fault plane properties. Polynomial chaos surrogates are built and validated to reproduce the uncertain Peak Ground Velocity (PGV), obtained from a discrete wavenumber/finite element method, at a set of 56 (virtual) stations. A sensitivity analysis is conducted to identify the main influent parameters: a partition of the uncertain input parameters into two groups highlights the strong impact of the hypocenter location. A Bayesian inference is then performed by using a Ground Motion Prediction Equation (GMPE) as observational measures. The results emphasize that additional physical constraints are valuable to increase the sampling efficiency.

[Figure]

The manuscript is clearly constructed and it would be suitable for the readership of the Geoscientific Model Development after the following revisions to clarify some aspects of the paper.

Specific comments

- page 6: one sentence is missing between line 4 and 5 to provide the number of terms $N_p$ in the PC series as a function of the stochastic space dimension $n_d$ and the total polynomial order $d$, $N_p = (d + n_d)!/(d!n_d!)$.

- page 6, line 19: the cross-validation process needs more details (leave-one-out or $k$-fold version, initial range of variation of the parameter $\gamma$ with the discretization strategy to find the optimal value) with a citation (e.g. the book of Seber and Lee, Linear regression analysis, 2003).

- page 7, section 3.1: the computation of the empirical error (8) with the training set $\mathcal{P}_{LHS}$ (blue dots) has only a minor interest because it simply shows that regression is a non-interpolating technique. A comparison between the empirical error estimated with the validation set (red dots) and a cross-validation error obtained with the training set is more relevant.

- page 8, line 12 (middle): the sentence "The overall tendency of PC prediction uncertainty (...) seems to decrease with increasing $R_{IJ}$ distance as well" relies on Fig. 6. This figure is hard to read and a new figure plotting only the (PC) standard deviations should be valuable (with a reminder in the text about the log-scale) to support the statement.

- page 8, line 16 (top): two stations are selected for plotting the PGV. Their locations must be indicated (for instance with labels on Fig. 2).

- page 8, line 1 (middle): The first sentence of the paragraph is incomplete since the complex dependency of PGVs to random inputs is not only due the mappings between the physical parameters and the standardized RVs $\{\xi_i\}_{1 \leq i \leq 7}$. We can speculate that the complexity of the propagation model (discrete wavenumber/finite element method) plays a major role.

- page 11: in Fig. 6, the GMPE standard deviation exhibits a higher level than the PC ones. A short discussion would be interesting to explain the causes/sources of this difference.

- page 13: a prediction error, defined as the discrepancy between the GMPE and PC series is introduced. This is confusing in Bayesian inference framework where observations (or measured data) are used to infer the model parameters. As GMPE predicted PGVs serve as observational data (see page 11), it would be more clear to replace GMPE by observational data (and to replace prediction error by observational error) in section 4.1.

Technical corrections

- page 2, line 9: replace is by are in "data is sufficient".

- page 2, line 16: replace Mw 65 by magnitude 65.

- page 5, Table 2, line 3: replace $y_h$ by $z_h$.

- page 6, line 18: "that" is missing, "note that $[\Psi]$ is station invariant".

- page 8, line 6 (top): the word "indeed" is useless.

[Figure]

Suggestions

- page 5, line 11: "number of stochastic dimensions" sounds weird. "stochastic space dimension" or "number of uncertain input parameters" are more usual.

- page 5, line 16: "instead of" seems to be inappropriate here and could be replaced by "which parameterize".

- page 6, line 13: the set of LHS realizations could be written, "...$N_{LHS} = 8000$ earthquake rupture model realizations through $\{\boldsymbol{\xi}_k\}_{1 \leq k \leq N_{LHS}}$".

- page 8, line 16: replace "with different PC truncation orders" by "with increasing odd PC truncation orders up to a degree nine".

- page 8, line 17: replace "PC library is sufficient ..." by "PC expansions are sufficiently accurate ...".

- pages 9 and 10: Fig. 4 and 5. represent distributions obtained with kernel density estimation. It should be mention in the captions or in the text.

- page 11, line 5: Move the group of words "for the same magnitude and focal mechanism" in section 3.2 (page 8), line 10 after the reference Boore and Atkinson (2008).

- page 13: explain a little bit more the partitioning of the data into four concentric groups (e.g. uniform discretization of the $R_{JB}$ interval).

- There is a huge number of ground motion predictions equations (see for example the report `http://www.gmpe.org.uk/gmpereport2014.pdf`). A short description of the GMPE model (for instance in an appendix) could be worthwhile to have a self-contained paper.

Please also note the supplement to this comment:
https://www.geosci-model-dev-discuss.net/gmd-2018-4/gmd-2018-4-RC3-supplement.pdf

—————————————————————————

---

## Editor Comment (EC1) · T. Poulet (Editor) · 21 Apr 2018

Dear authors,

Please note GMD's strong preference for the code to be uploaded as a supplement or to be made available at a data repository with an associated DOI (digital object identifier) for the exact model version described in the paper. Could you please add this modification to your manuscript?

Regards,

Thomas Poulet.

---

## Author Comment (AC1) · 16 May 2018

We thank the editor and all referees for providing us all the comments and feedback. We address all the comments below, and provide the revised manuscript as an attachment/supplement (in which all changes since our original manuscript have been highlighted in blue).

\*\*\*\* Editor's comment \*\*\*\*

(EC) »» Please note GMD's strong preference for the code to be uploaded as a supplement or to be made available at a data repository with an associated DOI (digital object identifier) for the exact model version described in the paper. Could you please add this modification to your manuscript?

(AC) »» We acknowledge the GMD's strong preserence for the code/data availability. The code used in this paper is originally developed by Spudich and Xu (2003) (see below). We are in the process of asking for permission from the authors to make the code publicly available. For now, we think it would make the most sense to maintain the following "Code and data availability" statement: The COMPSYN code (Spudich and Xu, 2003) employed in this study, along with the simulation data are available upon request.

In addition, we point to the online manual for the code at: https://www.researchgate.net/publication/260423574_Documentation_of_Software_Package_Compsyn_sxv311_Program_D_Green's_Functions

Spudich, P. and Xu, L.: 85.14-Software for Calculating Earthquake Ground Motions from Finite Faults in Vertically Varying Media, International Geophysics, 81, 1633–1634, 2003.

**** Referee 1 ****

(RC) »» page 6: one sentence is missing between line 4 and 5 to provide the number of terms Np in the PC series as a function of the stochastic space dimension nd and the total polynomial order d, Np = (d + nd)!/(d!nd!).

(AC) »» As suggested by the referee, the revised manuscript specifies the truncation strategy and provides an explicit formula for the size of the truncated basis.

(RC) »» page 6, line 19: the cross-validation process needs more details (leave-one-out or k-fold version, initial range of variation of the parameter $\gamma$ with the discretization strategy to find the optimal value) with a citation (e.g. the book of Seber and Lee,Linear regression analysis, 2003).

(AC) »» We used k-fold cross-validation (k=5) to determine the optimal $\gamma$. As suggested by the referee, the manuscript has been revised to provide details concerning the determination of the optimal $\gamma$ value. In addition, reference to the suggested citation has

been incorporated.

(RC) »» page 7, section 3.1: the computation of the empirical error (8) with the training set PLHS (blue dots) has only a minor interest because it simply shows that regression is a non-interpolating technique. A comparison between the empirical error estimated with the validation set (red dots) and a cross-validation error obtained with the training set is more relevant.

(AC) »» In our analysis of representation errors, we have examined both the cross-validation error, as well as the empirical error estimated using the training set, and have observed that the two error estimates are close to each other. A statement highlighting this observation has been added in the revised manuscript (specifically the caption of Fig. 3).

(RC) »» page 8, line 12 (middle): the sentence "The overall tendency of PC prediction uncertainty (...) seems to decrease with increasing RIJ distance as well" relies on Fig. 6. This figure is hard to read and a new figure plotting only the (PC) standard deviations should be valuable (with a reminder in the text about the log-scale) to support the statement.

(AC) »» As suggested by the referee, we have attempted to plot the PC standard deviations independently, but this did not lead to dramatic improvement in the presentation, namely because the distant stations are clustered (in Rjb distance measure). On the other hand, the referee's suggestion concerning the log-scale has been incorporated (caption of Fig. 6).

(RC) »» page 8, line 16 (top): two stations are selected for plotting the PGV. Their locations must be indicated (for instance with labels on Fig. 2).

(AC) »» The referee's comment has been implemented. (See Fig. 2) Note, in the revised manuscript, we decided to show PC statistics on Station #3 and #22 (instead of #3 and #21). The reason for this switch is the following: Station #21 turns out to

be very close to station #3. To better illustrate the validity of our PC surrogates over a distance, we decided to select a station (#22) that is a bit far from station #3. (Fig. 4 and Fig. 5 are updated accordingly.)

(RC) »» page 8, line 12 (middle): The first sentence of the paragraph is incomplete since the complex dependency of PGVs to random inputs is not only due the mappings between the physical parameters and the standardized RVs f$\xi$ig1≤i≤7. We can speculate that the complexity of the propagation model (discrete wavenumber/finite element method) plays a major role.

(AC) »» We agree with the referee that the sentence in question is confusing. Our intention was to highlight that the conditional mapping between canonical rv's and physical parameters makes it difficult to isolate the impact of individual parameters, but that this difficulty can be effectively addressed using global sensitivity analysis. The manuscript has been revised to clarify this aspect.

(RC) »» page 11: in Fig. 6, the GMPE standard deviation exhibits a higher level than the PC ones. A short discussion would be interesting to explain the causes/sources of this difference.

(AC) »» It turned out that in our original Fig. 6, we have plotted 2 times the GMPE standard deviation bounds. We apologize for the confusion, and have updated the Fig. 6 with one standard deviation GMPE bounds. The new Fig.6 shows similar standard deviation bounds between GMPE and PC statistics in general. However, one should not expect exact match between GMPE and PC statistics, due to difference in random sources underlying the two approaches, and the uninformative PC random variable distribution used to calculate the statistics.

(RC) »» page 13: a prediction error, defined as the discrepancy between the GMPE and PC series is introduced. This is confusing in Bayesian inference framework where observations (or measured data) are used to infer the model parameters. As GMPE predicted PGVs serve as observational data (see page 11), it would be more clear to

Interactive
comment

replace GMPE by observational data (and to replace prediction error by observational error) in section 4.1.

(AC) »» We agree with the referee's comments. The manuscript has been revised accordingly.

Technical Corrections

(RC) »» page 2, line 9: replace is by are in "data is sufficient". page 2, line 16: replace Mw 6.5 by magnitude 6.5. page 5, Table 2, line 3: replace yh by zh. page 6, line 18: "that" is missing, "note that [Ψ] is station invariant". page 8, line 6 (top): the word "indeed" is useless.

(AC) »» The suggested corrections above have been implemented.

Suggestions

(RC) »» page 5, line 11: "number of stochastic dimensions" sounds weird. "stochastic space dimension" or "number of uncertain input parameters" are more usual

(AC) »» As suggested by the referee, we replaced "number of stochastic dimensions" with "stochastic space dimension"

(RC) »» page 5, line 16: "instead of" seems to be inappropriate here and could be replaced by "which parameterize". page 6, line 13: the set of LHS realizations could be written, "... NLHS = 8000 earthquake rupture model realizations through f$\xi$kg1$\leq$k$\leq$NLHS". page 8, line 16: replace "with different PC truncation orders" by "with increasing odd PC truncation orders up to a degree nine". page 8, line 17: replace "PC library is sufficient ..." by "PC expansions are sufficiently accurate ...".

(AC) »» The suggestions above have been implemented.

(RC) »» pages 9 and 10: Fig. 4 and 5. represent distributions obtained with kernel density estimation. It should be mention in the captions or in the text.

**[GMDD](GMDD)**
[Figure]

(AC) »» The captions of Figs. 4 and 5 have been modified as suggested.

(RC) »» page 11, line 5: Move the group of words "for the same magnitude and focal mechanism" in section 3.2 (page 8), line 10 after the reference Boore and Atkinson (2008).

(AC) »» This suggestion has been implemented.

(RC) »» page 13: explain a little bit more the partitioning of the data into four concentric groups (e.g. uniform discretization of the RJB interval).

(AC) »» As suggested by the referee, additional details have been added to the manuscript to explain the partitioning of the data into four concentric group. This partition is motivated by the observation of PGV variability decaying with Rjb distance (Figure 6), and is to ensure that the inference appropriately accounts for different PGV variance at different Rjb distances. (The 4-group partition criterion is added to the legend of Fig. 2).

(RC) »» There is a huge number of ground motion predictions equations (see for example the report http://www.gmpe.org.uk/gmpereport2014.pdf). A short description of the GMPE model (for instance in an appendix) could be worthwhile to have a self-contained paper.

(AC) »» In addition to the original reference, the GMPE model [BA2008] used has been discussed in a number of accessible references, which have been incorporated in the revised manuscript, more specifically, the following three resources have been added in the manuscript (footnote in the discussion of Fig. 6): http://www.opensha.org/glossary-attenuationRelation-BOORE\_ATKIN\_2008 http://www.gmpe.org.uk/gmpereport2014.pdf Mai (2009) Consequently, we feel that addition of an Appendix is not necessary, and may dilute the focus of the work.

\*\*\*\* Referee #2 \*\*\*\*

(RC) »» The authors develop a polynomial chaos (PC) expansion representation to provide a surrogate model for a probability distribution of Mw 6.5 strike-slip earthquakes with a fixed fault geometry. Seven parameters are used to describe a particular realization, including the hypocenter location and parameters describing an elliptical asperity, a region of relatively high slip, defining a 7-dimensional stochastic space. The surrogate model allows the rapid estimation of the peak ground velocity (PGV) at each of 56 virtual observation points. The PC expansion is computed using synthetic seismogram observations at these points for a set of 8000 realizations. A second set of 8000 realizations is used for validation, to confirm that the surrogate model constructed from the first set agrees well with the direct simulation results for the second set of realizations.The surrogate model is then used to rapidly compute the PGV for millions of additional realizations in order to gather statistics on the decay of PGV with respect to distance from the fault (measured using the Joyner-Boore distance RJB, the minimal distance to the fault plane as projected to the surface), at the 56 observation points. The mean PGV and standard deviation at each observation point are plotted vs. the distance RJB, and this data compared with the ground motion prediction equation (GMPE) of Boore and Atkinson (2008). The GMPE was derived based on observations of past earthquakes and so it is interesting to see that the statistics generated by the PC expansion generally follows this prediction and lie within one standard deviation of the GMPE as determined by Boore and Atkinson. This suggests that a simplified fault model consisting of a single asperity and a small set of parameters can perhaps predict PGV statistics well, and hence may be useful for predicting other GMPE curves, or for probabilistic seismic hazard analysis more generally. The first 3 sections of the paper give a nice development of these ideas.

I had more trouble understanding the goal of Section 4, which concerns the use of Bayesian inference to determine a probability distribution on the space of PC parameters that yield an event to best match the GMPE. It seems to me that the GMPE is only intended to predict the average and standard deviation of the PGV over a large set of potential earthquakes, and so I do not understand the point of this statistical

inversion to try to determine the characteristics of one particular earthquake that best matches the average. The authors conclude that the best match is more likely to have the hypocenter located in the lower right quadrant of the fault plane, and the elliptical patch centered in the lower left quadrant. Why is this useful to know? Is this meant to have geophysical significance, e.g. that real strike-slip earthquakes of this magnitude tend to have their hypocenter and asperities located in this way? How does this relate to the actual slip patterns of the real events that went into the Boore and Atkinson GMPE model, to the extent those are known? There is no discussion in the paper of these topics. I also wonder about the way this inversion is used in Section 4.5, as discussed in one of my specific comments below. I think the paper would be stronger if the motivation for doing this inversion was better explained, since I found it hard to assess the usefulness of this part of the paper.

(AC) »» The referee stated: "It seems to me that the GMPE is only intended to predict the average and standard deviation of the PGV over a large set of potential earthquakes, and so I do not understand the point of this statistical inversion to try to determine the characteristics of one particular earthquake that best matches the average." However, this interpretation is incorrect. This paper focus on the class of earthquakes of magnitude M=6.5 with strike slip focal mechanism. It is true that GMPE predictions for the same class of earthquakes are statistical averages over many earthquakes and regions, the amount of available data for GMPE predictions are still sparse. On the other hand, this paper aimed at exploring the capability of our PC approach in reproducing ground-motions of the same class of earthquake; and our rupture model simulations and PC analyses show that we don't need such GMPE in principle.

The referee expressed his/her concern in understanding the conclusion of "the best match is more likely to have the hypocenter located in the lower right quadrant of the fault plane, and the elliptical patch centered in the lower left quadrant." We point out that this particular interpretation/conclusion (hypocenter on the right while elliptical patch on the left of the fault plane) results from the station distribution; if we had put an

exactly regular/symmetric station distribution, the patch could also be in the right and the hypocenter in the left. The important message here is that hypocenter and slip patch cannot be in near-surface area of the fault, and they need to have some distance from each other in order to produce the proper seismic radiation pattern, including on-fault directivity. Otherwise, the near-source waveforms, and hence PGVs, would not match. This is consistent with the findings of Mai et al (2005).

The referee raised more follow up questions in understanding our conclusions on the most likely fault plane configuration, e. g. "Why is this useful to know? Is this meant to have geophysical significance, e.g. that real strike-slip earthquakes of this magnitude tend to have their hypocenter and asperities located in this way? How does this relate to the actual slip patterns of the real events that went into the Boore and Atkinson GMPE model, to the extent those are known?" We would like to point out that the GMPE (BA2008) relations are based on many earthquakes. Unfortunately, there exist no such detailed source information (i.e. fault plane configuration as considered in our paper) for most of those earthquakes. Furthermore, the GMPE (BA2008) relations do not parameterize any of the source complexity considered in our paper. The important message again is that our finding is backed up by independent observations and physical arguments in Mai et al (2005).

Revision has been made to clarify our main conclusions in the conclusion section.

(RC) »» Page 3, line 2: The fault plane geometry is fixed with width 10 km and length 27 km. It is stated that this is obtained from 100 realizations following the scaling relation in Wells and Coppersmith (1994). How are 100 realizations used to determine these dimensions?

(AC) »» Following scaling relations, e.g. Wells and Coppersmith (1994), Mai and Beroza (2000) and Thingbaijam et al (2017), we obtained 100 possible values of rupture lengths for a M 6.5 strike-slip event and found that L=27 km had the highest population in our histogram. We did the same for the rupture width and obtained W=10km.

Revision has been implemented to clarify our choice of the fault plane width and length.

(RC) »» Page 3, lines 5–7: Why is the slip set to Smax/e outside the asperity? How is the slip in the asperity set? Since the area of the asperity varies with the input parameters, the slip must also vary to keep the magnitude fixed. It is stated that Smax varies with the ellipse size but it is not clear how.

(AC) »» We noticed that the referee might misunderstand our description about the way we set the slip in the whole fault plane. For the slip inside the asperity, we state that "the ellipse is the asperity with Gaussian slip distribution inside". We pointed out in the manuscript that "The maximum slip Smax is chosen such that the mean slip remains constant (0.71 m) when varying the ellipse size." It is important to note that the the moment magnitude Mw depends on the mean slip of the whole fault plane, and not only from the slip of the area of the asperity. The slip between the elliptical patch boundary and dashed rectangle is set to Smax/e, the minimum value at the patch boundary from the Gaussian slip distribution;

(RC) »» Page 3, line 15–17: For completeness it would be good to state the grid resolution used in the COMPSYN simulation of the seismic signals, and the domain size, boundary conditions imposed, etc.

(AC) »» As suggested by the referee, the following details have been added to our revision:

COMPSYN solves the equation of motion considering initial conditions of zero displacement and velocity at a reference time t0 and specifying traction or displacement on the bounding surface of the medium (boundary conditions) using the unit outward normal vector (details about the scheme can be seen in Olson et al., 1984). The grid resolution used in COMPSYN is variable and uses a spacing of 1/6 of the minimum shear wavelength at depth z. The grid extends a total depth that depends on the wavenumber, which means that the maximum depth decreases when the wavenumber increases.

Interactive
comment

(RC) »» Page 3, Figure 2: The 56 observation stations surround the fault plane on all sides. Since the fault plane is vertical and the velocity model is vertically layered, shouldn't the observations be symmetric about Y = 0? If so, it would seem clearer to simply use points in the upper half plane, for example, rather than asymmetric points scattered on both sides.

(AC) »» We thank the referee for this important observation. In principle this observation is correct, and it is possible to use points in the upper half plane only, as pointed out by the referee, however the stations are not exactly symmetrically arranged, for the very reason to somewhat disturb the symmetry of the problem.

(RC) »» Page 11, Figure 6: The points here are presumably the mean PGV observed at each of the 56 observation points, plotted vs. the distance RJB. These points are calculated by evaluating the PC expansion at 1,000,000 sample points and are presumably quite accurate estimates of the mean at each observation point. But this figure shows that two points that have very similar RJB can have quite different PGV, presumably because the two points have quite different azimuthal orientation relative to the fault, even though they are the same distance away. This is interesting to observe, but since the GMPE curve ignores orientation it seems like it might also be interesting to try to average over different orientations for each distance. This could be facilitated if a number of observation points were placed at each distance, for a discrete set of distances, i.e., place the observation points on concentric rings with fixed RJB. It also seems like a much larger set of observation points could be used than 56, since the PC model is so quick to evaluate. If many points were placed on many different concentric rings, then one could average over all points at a given distance to get points that might be expected to agree better with the GMPE curve in Figure 6. It would then also be possible to explore in more detail how the PGV varies with orientation along each ring.

(AC) »» This is a very good observation. We thought about this already: variations in PGV at a given distance are likely due to radiation-pattern effects, in particular directivity. As pointed out by the reviewer, one could now do many more detailed tests,

including using the PC approach to explore the ground motion dependency on azimuthal orientation. However, it would require the construction and validation of additional PC representations for a large number of observation stations, which are beyond the scope of this study (i.e. the first of its kind to apply PC-expansion to ground-motion prediction).

Instead we refer to recently published study by Vyas et al (2016) that exactly addresses this question in great detail, with a range of simulations and 3000 randomly distributed sites.

Vyas, J. M. Galis, and P. M. Mai (2016). Distance and azimuthal dependence of ground-motion variability, Bull. Seis. Soc. Am. Vol. 106, No. 4, doi: 10.1785/0120150298.

The following sentences have been added to the discussion of Fig. 6 in our revised manuscript. "It is noted that two stations with similar Rjb distance can have very different PGV values. This is likely due to radiation-pattern effects, in particular directivity, which is addressed in great details by Vyas et al (2016)."

(RC) »» In Figure 2 there are sets of points that have different colors/symbols that are arranged somewhat in rings, but the distance for each color do not seem to be constant. The use of colors/symbols is not explained anywhere I could find, and should be.

(AC) »» We have updated Figure 2 to provide details concerning the grouping of observation states into four concentric sets, and to indicate that the color/symbols are used to highlight this grouping. In addition, we also indicate the locations of two selected stations in Figure 4 and 5.

(RC) »» Page 5, Table 2: The caption says that "(*) denotes dependent parameters". It is not clear what this means. Does this refer to the comment in line 5 of this page, where it is noted that "These restrictions lead to nonlinear dependency between feasible ranges of different physical parameters"?

(AC) »» In the revised manuscript, we have modified the caption of Figure 2 as follows: "Parameters governing fault plane configurations, (*) denotes parameters whose feasible ranges are dependent on others."

(RC) »» The fact that some of these parameters are constrained based on the choice of other parameters means that the probability distribution of parameters is not really given by (1) on page 5 as is stated. Some choices from this 7-dimensional box have probability zero due to the constraints, while others have greater probability due to several non-allowed choices mapping to the same set of modified parameters when the asperity falls near the edge of the fault plane. Does this affect the validity of the PC expansion and/or results? At any rate, this should be discussed.

(AC) »» A brief discussion has been added in the revised manuscript (beginning of section 3) in order to highlight the distinction between canonical random variables, which are iid uniform over the 7-dimensional hypercube, and physical parameters whose ranges may be interdependent. The PC expansion is constructed in terms of the canonical random variables, and its validity is tested using cross-validation and empirical error estimates.

(RC) »» Page 17, Section 4.5: In this section it is stated that a uniform distribution of parameters over the 7-dimensional space ignores various geophysical constraints suggested by previous work. This is discussed in the context of choosing a prior for the Bayesian inference, but it seems like it would be even more important to incorporate these constraints into the analysis of Section 3, where the PC expansion is used to generate statistics on the PGV for comparison with the GMPE. Why should the statistics obtained with the uniform distribution be expected to match the GMPE well if it is known that this is the wrong distribution? This is addressed to some extent in Section 4.5 where the inversion that incorporates these constraints is then used to generate statistics that are compared to the GMPE curve in Figure 15. But at this point the inversion process has been used to to further constrain the posterior distribution based on trying to match the GMPE curve, so comparing the result to the GMPE curve does not seem to provide any validation that the PC expansion could predict the GMPE curve
for other scenarios, for example. I may be missing the point here, but I think it needs more explanation.

(AC) »» 1. "Why should the statistics obtained with the uniform distribution be expected to match the GMPE well if it is known that this is the wrong distribution?" The PC statistics and GMPE results were compared to ensure that the model predictions describe a similar range, which consequently enables us to use the GMPE results as "data" for the purpose of parameter inference. Without this, it wouldn't be reasonable to use GMPE reference curve as "observation" in the Bayesian framework. 2. "so comparing the result to the GMPE curve does not seem to provide any validation that the PC expansion could predict the GMPE curve for other scenarios" As pointed out earlier, the PC expansion was designed to provide an efficient representation of the model behavior. In building the PC representation, we relied on uninformative prior, that spans a wide range of feasible scenarios. In Section 3, we verified the capability of the PC surrogate in reproducing the model predictions over the considered parameter ranges. As discussed in Alexanderian et al. (2012), one of the advantages of having a suitable representation over a wide range of parameters is that the restriction of parameter ranges can be efficiently performed a posteriori, namely without the need of performing new model simulations. This advantage was specifically exploited in section 4.

As suggested by the referee, additional explanation has been incorporated in the revised manuscript concerning the construction and validation of the PC expansion, and later on the restrictions explored in the Bayesian analysis.

Technical Corrections

(RC) »» Page 3, line 2: Presumably the rake is fixed at 0 degrees for a strike-slip event, but this should perhaps be stated?

(AC) »» As pointed out by the referee, the rake value has been added in the revised manuscript.

[Figure]

(RC) »» Page 7, line 27: What are the index sets Si and Ti? The sets are used in the summations of (7a) and (7b) respectively, but not really defined.

(AC) »» The definitions of Si and Ti have been included in the revised manuscript.

(RC) »» Proper latex fonts for trig functions should be used in expressions such as (A1), e.g. a cos $\beta$ rather than a cos $\beta$.

(AC) »» This comment has been incorporated. (not highlighted in the revised manuscript, as it is quite trivial)

Please also note the supplement to this comment:
https://www.geosci-model-dev-discuss.net/gmd-2018-4/gmd-2018-4-AC1-supplement.pdf

———————————————

[Figure]

**Supplement:**

**Bayesian inference of earthquake rupture models using polynomial chaos expansion**

Hugo Cruz-Jiménez[1], Guotu Li[2], Paul Martin Mai[1], Ibrahim Hoteit[1], and Omar M. Knio[1,2]

[1]King Abdullah University of Science and Technology, Thuwal 23955, Saudi Arabia
[2]Duke University, Durham, NC 27708, USA

*Correspondence to:* Guotu Li (guotu.li@duke.edu); Omar M. Knio (Omar.Knio@kaust.edu.sa);

**Abstract.** In this paper we employed polynomial chaos (PC) expansions to understand earthquake rupture model responses to random fault plane properties. A sensitivity analysis based on our PC surrogate model suggests that the hypocenter location plays a dominant role in peak ground velocity (PGV) responses, while elliptical patch properties only show secondary impact.

5  In addition, the PC surrogate model is utilized for Bayesian inference of the most likely underlying fault plane configuration in light of a set of PGV observations from a ground motion prediction equation (GMPE). A restricted sampling approach is also developed to incorporate additional physical constraints on the fault plane configuration, and to increase the sampling efficiency.

**Keywords.** Polynomial Chaos expansion, Sensitivity analysis, Bayesian inference, Earthquake seismology, Peak ground ve-

10  locity.

*Copyright statement.* © Author(s) 2017. This work is distributed under the Creative Commons Attribution 4.0 License.

[revised manuscript text omitted]

---

## Author Response (AR1)

**Summary of Modifications to Manuscript GMD-2018-4**

**Bayesian Inference of Earthquake Rupture Models Using Polynomial Chaos Expansion**

Hugo Cruz-Jiménez, Guotu Li Paul Martin Mai, Ibrahim Hoteit Omar M. Knio

We would like to thank the editor and referees for their thorough reviews, comments and suggestions. Below is a summary of modifications to the original manuscript, in response to the editor and referees' comments/suggestions.

We hope that with these modifications, the present manuscript will be found suitable for publication in *Geoscientific Model Development*.

**Editor's comment**

Please note GMD's strong preference for the code to be uploaded as a supplement or to be made available at a data repository with an associated DOI (digital object identifier) for the exact model version described in the paper. Could you please add this modification to your manuscript?

**Reply:**

The COMPSYN code used in this paper was originally developed by Spudich and Xu (2003) (see below). We are in the process of seeking permission from the authors to make the code publicly available. We hope to make the code available on a dedicated site/repository in the near future (as stated in the revised manuscript).

In addition, we point to the online manual for the code at:

https://www.researchgate.net/publication/260423574\_Documentation\_of\_Software\_Package\_C ompsyn\_sxv311\_Programs\_for\_Earthquake\_Ground\_Motion\_Calculation\_Using\_Complete\_1-D\_Green's\_Functions.

Regarding the code for polynomial chaos expansion framework, we have added additional references, as well as a link to an open source toolkit available at http://www.sandia.gov/UQToolkit/

**Referee 1**

**General comments**

This manuscript investigates an earthquake rupture model subject to 7 random fault plane properties. Polynomial chaos surrogates are built and validated to reproduce the uncertain Peak Ground Velocity (PGV), obtained from a discrete wavenumber/finite element method, at a set of 56 (virtual) stations. A sensitivity analysis is conducted to identify the main influent parameters: a partition of the uncertain input parameters into two groups highlights the strong impact of the hypocenter location. A Bayesian inference is then performed by using a Ground Motion Prediction Equation (GMPE) as observational measures. The results emphasize that additional physical constraints are valuable to increase the sampling efficiency.

The manuscript is clearly constructed and it would be suitable for the readership of the Geoscientific Model Development after the following revisions to clarify some aspects of the paper.

**Specific comments**

 page 6: one sentence is missing between line 4 and 5 to provide the number of terms Np in the PC series as a function of the stochastic space dimension nd and the total polynomial order d, Np = (d + nd)!/(d!nd!).

**Reply:** As suggested by the referee, the revised manuscript specifies the truncation strategy and provides an explicit formula for the size of the truncated basis.

page 6, line 19: the cross-validation process needs more details (leave-one-out or k-fold version, initial range of variation of the parameter γ with the discretization strategy to find the optimal value) with a citation (e.g. the book of Seber and Lee,Linear regression analysis, 2003).

**Reply:** We used k-fold (k=5) cross-validation to determine the optimal  $\gamma$ . As suggested by the referee, the manuscript has been revised to provide details concerning the determination of the optimal  $\gamma$  value. In addition, reference to the suggested citation has been incorporated.

• page 7, section 3.1: the computation of the empirical error (8) with the training set PLHS (blue dots) has only a minor interest because it simply shows that regression is a non-interpolating technique. A comparison between the empirical error estimated with the validation set (red dots) and a cross-validation error obtained with the training set is more relevant.

**Reply:** In our analysis of representation errors, we have examined both the cross-validation error, as well as the empirical error estimated using the training set, and have observed that the

two error estimates are close to each other. A statement highlighting this observation has been added in the revised manuscript (specifically the caption of Fig. 3).

 page 8, line 12 (middle): the sentence "The overall tendency of PC prediction uncertainty (...) seems to decrease with increasing RIJ distance as well" relies on Fig. 6. This figure is hard to read and a new figure plotting only the (PC) standard deviations should be valuable (with a reminder in the text about the log-scale) to support the statement.

**Reply:** As suggested by the referee, we have attempted to plot the PC standard deviations independently, but this did not lead to dramatic improvement in the presentation, namely because the distant stations are clustered (in Rjb distance measure). On the other hand, the referee's suggestion concerning the log-scale has been incorporated (caption of Fig. 6).

• page 8, line 16 (top): two stations are selected for plotting the PGV. Their locations must be indicated (for instance with labels on Fig. 2).

**Reply:** The referee's comment has been implemented. (See Fig. 2) Note, in the revised manuscript, we decided to show PC statistics on Station #3 and #22 (instead of #3 and #21 in the original manuscript). The reason for this switch is the following: Station #21 turns out to be very close to station #3. To better illustrate the validity of our PC surrogates over a distance, we decided to select a station (#22) that is a bit far from station #3. (Fig. 4 and Fig. 5 are updated accordingly.)

 page 8, line 12 (middle): The first sentence of the paragraph is incomplete since the complex dependency of PGVs to random inputs is not only due the mappings between the physical parameters and the standardized RVs fξig1≤i≤7. We can speculate that the complexity of the propagation model (discrete wavenumber/finite element method) plays a major role.

**Reply:** We agree with the referee that the sentence in question is confusing. Our intention was to highlight that the conditional mapping between canonical rv's and physical parameters makes it difficult to isolate the impact of individual parameters, but that this difficulty can be effectively addressed using global sensitivity analysis. The manuscript has been revised to clarify this aspect.

• page 11: in Fig. 6, the GMPE standard deviation exhibits a higher level than the PC ones. A short discussion would be interesting to explain the causes/sources of this difference.

**Reply:** It turned out that in our original Fig. 6, we have plotted 2 times the GMPE standard deviation bounds. We apologize for the confusion, and have updated the Fig. 6 with one standard deviation GMPE bounds. The new Fig.6 shows similar standard deviation bounds

between GMPE and PC statistics in general. However, one should not expect exact match between GMPE and PC statistics, due to difference in random sources underlying the two approaches, and the uninformative PC random variable distribution used to calculate the statistics.

• page 13: a prediction error, defined as the discrepancy between the GMPE and PC series is introduced. This is confusing in Bayesian inference framework where observations (or measured data) are used to infer the model parameters. As GMPE predicted PGVs serve as observational data (see page 11), it would be more clear to replace GMPE by observational data (and to replace prediction error by observational error) in section 4.1.

**Reply:** We agree with the referee's comments. The manuscript has been revised accordingly.

**Technical Corrections**

- page 2, line 9: replace is by are in "data is sufficient".
- page 2, line 16: replace Mw 6.5 by magnitude 6.5.
- page 5, Table 2, line 3: replace yh by zh.
- page 6, line 18: "that" is missing, "note that [Ψ] is station invariant".
- page 8, line 6 (top): the word "indeed" is useless.

**Reply:** The suggested corrections above have been implemented in the revised manuscript.

**Suggestions**

• page 5, line 11: "number of stochastic dimensions" sounds weird. "stochastic space dimension" or "number of uncertain input parameters" are more usual

**Reply:** As suggested by the referee, we replaced "number of stochastic dimensions" with "stochastic space dimension"

- page 5, line 16: "instead of" seems to be inappropriate here and could be replaced by "which parameterize".
- page 6, line 13: the set of LHS realizations could be written, "... NLHS = 8000 earthquake rupture model realizations through fξkg1≤k≤NLHS".
- page 8, line 16: replace "with different PC truncation orders" by "with increasing odd PC truncation orders up to a degree nine".
- page 8, line 17: replace "PC library is sufficient ..." by "PC expansions are sufficiently accurate ...".

**Reply:** The suggestions above have been implemented in the revised manuscript.

• pages 9 and 10: Fig. 4 and 5. represent distributions obtained with kernel density estimation. It should be mention in the captions or in the text.

**Reply:** The captions of Figs. 4 and 5 have been modified as suggested.

• page 11, line 5: Move the group of words "for the same magnitude and focal mechanism" in section 3.2 (page 8), line 10 after the reference Boore and Atkinson (2008).

**Reply:** This suggestion has been implemented.

• page 13: explain a little bit more the partitioning of the data into four concentric groups (e.g. uniform discretization of the RJB interval).

**Reply:** As suggested by the referee, additional details have been added to the revised manuscript to explain the partitioning of the data into four groups. This partition is motivated by the observation of PGV variability decaying with Rjb distance (Figure 6), and is to ensure that the inference appropriately accounts for different PGV variance at different Rjb distances. (The 4-group partition criterion is added to the legend of Fig. 2).

• There is a huge number of ground motion predictions equations (see for example the report http://www.gmpe.org.uk/gmpereport2014.pdf). A short description of the GMPE model (for instance in an appendix) could be worthwhile to have a self-contained paper.

**Reply:** In addition to the original reference, the GMPE model [BA2008] used has been discussed in a number of accessible references, which have been incorporated in the revised manuscript, more specifically, the following three resources have been added in the revised manuscript (footnote in the discussion of Fig. 6):

- 1) http://www.opensha.org/glossary-attenuationRelation-BOORE\\_ATKIN\\_2008
- 2) http://www.gmpe.org.uk/gmpereport2014.pdf
- 3) Mai (2009)

Consequently, we feel that addition of an Appendix is not necessary, and may dilute the focus of the work.

**Referee #2**

**General comments**

The authors develop a polynomial chaos (PC) expansion representation to provide a surrogate model for a probability distribution of Mw 6.5 strike-slip earthquakes with a fixed fault geometry. Seven parameters are used to describe a particular realization, including the hypocenter location and parameters describing an elliptical asperity, a region of relatively high slip, defining a 7-dimensional stochastic space. The surrogate model allows the rapid estimation of the peak ground velocity (PGV) at each of 56 virtual observation points. The PC expansion is computed using synthetic seismogram observations at these points for a set of 8000 realizations. A second set of 8000 realizations is used for validation, to confirm that the surrogate model constructed from the first set agrees well with the direct simulation results for the second set of realizations. The surrogate model is then used to rapidly compute the PGV for millions of additional realizations in order to gather statistics on the decay of PGV with respect to distance from the fault (measured using the Joyner-Boore distance RJB, the minimal distance to the fault plane as projected to the surface), at the 56 observation points. The mean PGV and standard deviation at each observation point are plotted vs. the distance RJB, and this data compared with the ground motion prediction equation (GMPE) of Boore and Atkinson (2008). The GMPE was derived based on observations of past earthquakes and so it is interesting to see that the statistics generated by the PC expansion generally follows this prediction and lie within one standard deviation of the GMPE as determined by Boore and Atkinson. This suggests that a simplified fault model consisting of a single asperity and a small set of parameters can perhaps predict PGV statistics well, and hence may be useful for predicting other GMPE curves, or for probabilistic seismic hazard analysis more generally. The first 3 sections of the paper give a nice development of these ideas.

I had more trouble understanding the goal of Section 4, which concerns the use of Bayesian inference to determine a probability distribution on the space of PC parameters that yield an event to best match the GMPE. It seems to me that the GMPE is only intended to predict the average and standard deviation of the PGV over a large set of potential earthquakes, and so I do not understand the point of this statistical inversion to try to determine the characteristics of one particular earthquake that best matches the average. The authors conclude that the best match is more likely to have the hypocenter located in the lower right quadrant of the fault plane, and the elliptical patch centered in the lower left quadrant. Why is this useful to know? Is this meant to have geophysical significance, e.g. that real strike-slip earthquakes of this magnitude tend to have their hypocenter and asperities located in this way? How does this relate to the actual slip patterns of the real events that went into the Boore and Atkinson GMPE model, to the extent those are known? There is no discussion in the paper of these topics. I also wonder about the way this inversion is used in Section 4.5, as discussed in one of my specific comments below. I think the paper would be stronger if the motivation for doing this inversion was better explained, since I found it hard to assess the usefulness of this part of the paper.

**Reply:**

- 1. The referee stated: "It seems to me that the GMPE is only intended to predict the average and standard deviation of the PGV over a large set of potential earthquakes, and so I do not understand the point of this statistical inversion to try to determine the characteristics of one particular earthquake that best matches the average." However, this interpretation is incorrect. This paper focus on the class of earthquakes of magnitude M=6.5 with strike slip focal mechanism. It is true that GMPE predictions for the same class of earthquakes are statistical averages over many earthquakes and regions, the amount of available data for GMPE predictions are still sparse. On the other hand, this paper aimed at exploring the capability of our PC approach in reproducing ground-motions of the same class of earthquake; and our rupture model simulations and PC analyses show that we don't need such GMPE in principle.
- 2. The referee expressed his/her concern in understanding the conclusion of "the best match is more likely to have the hypocenter located in the lower right quadrant of the fault plane, and the elliptical patch centered in the lower left quadrant." We point out that this particular interpretation/conclusion (hypocenter on the right while elliptical patch on the left of the fault plane) results from the station distribution; if we had put an exactly regular/symmetric station distribution, the patch could also be in the right and the hypocenter in the left. The important message here is that hypocenter and slip patch cannot be in near-surface area of the fault, and they need to have some distance from each other in order to produce the proper seismic radiation pattern, including on-fault directivity. Otherwise, the near-source waveforms, and hence PGVs, would not match. This is consistent with the findings of Mai et al (2005).
- 3. The referee raised more follow up questions in understanding our conclusions on the most likely fault plane configuration, e. g. "Why is this useful to know? Is this meant to have geophysical significance, e.g. that real strike-slip earthquakes of this magnitude tend to have their hypocenter and asperities located in this way? How does this relate to the actual slip patterns of the real events that went into the Boore and Atkinson GMPE model, to the extent those are known?" We would like to point out that the GMPE (BA2008) relations are based on many earthquakes. Unfortunately, there exist no such detailed source information (i.e. fault plane configuration as considered in our paper) for most of those earthquakes. Furthermore, the GMPE (BA2008) relations do not parameterize any of the source complexity considered in our paper. The important message again is that our finding is backed up by independent observations and physical arguments in Mai et al (2005).
- 4. Revision has been made to clarify our main conclusions in the conclusion section.

**Specific comments**

• Page 3, line 2: The fault plane geometry is fixed with width 10 km and length 27 km. It is stated that this is obtained from 100 realizations following the scaling relation in Wells and Coppersmith (1994). How are 100 realizations used to determine these dimensions?

**Reply:** Following scaling relations, e.g. Wells and Coppersmith (1994), Mai and Beroza (2000) and Thingbaijam et al (2017), we obtained 100 possible values of rupture lengths for a M=6.5 strike-slip event and found that L=27 km had the highest population in our histogram. We did the same for the rupture width and obtained W=10km. Revision has been implemented to clarify our choice of the fault plane width and length.

• Page 3, lines 5–7: Why is the slip set to Smax/e outside the asperity? How is the slip in the asperity set? Since the area of the asperity varies with the input parameters, the slip must also vary to keep the magnitude fixed. It is stated that Smax varies with the ellipse size but it is not clear how.

**Reply:** We noticed that the referee might misunderstand our description about the way we set the slip in the whole fault plane.

- 1) For the slip inside the asperity, we state that "the ellipse is the asperity with Gaussian slip distribution inside".
- 2) We pointed out in the manuscript that "The maximum slip Smax is chosen such that the mean slip remains constant (0.71 m) when varying the ellipse size." It is important to note that the moment magnitude Mw depends on the **mean slip of the whole fault plane**, and not only from the slip of the area of the asperity.
- 3) The slip between the elliptical patch boundary and dashed rectangle is set to Smax/e, the minimum value at the patch boundary from the Gaussian slip distribution;
- Page 3, line 15–17: For completeness it would be good to state the grid resolution used in the COMPSYN simulation of the seismic signals, and the domain size, boundary conditions imposed, etc.

**Reply:** As suggested by the referee, the following details have been added to our revision:

COMPSYN solves the equation of motion considering initial conditions of zero displacement and velocity at a reference time t0 and specifying traction or displacement on the bounding surface of the medium (boundary conditions) using the unit outward normal vector (details about the scheme can be seen in Olson et al., 1984). The grid resolution used in COMPSYN is variable and uses a spacing of 1/6 of the minimum shear wavelength at depth z. The grid extends a total depth that depends on the wavenumber, which means that the maximum depth decreases when the wavenumber increases.

 Page 3, Figure 2: The 56 observation stations surround the fault plane on all sides. Since the fault plane is vertical and the velocity model is vertically layered, shouldn't the observations be symmetric about Y = 0? If so, it would seem clearer to simply use points in the upper half plane, for example, rather than asymmetric points scattered on both sides. **Reply:** We thank the referee for this important observation. In principle this observation is correct, and it is possible to use points in the upper half plane only, as pointed out by the referee, however the stations are not exactly symmetrically arranged, for the very reason to somewhat disturb the symmetry of the problem.

• Page 11, Figure 6: The points here are presumably the mean PGV observed at each of the 56 observation points, plotted vs. the distance RJB. These points are calculated by evaluating the PC expansion at 1,000,000 sample points and are presumably quite accurate estimates of the mean at each observation point. But this figure shows that two points that have very similar RJB can have guite different PGV, presumably because the two points have quite different azimuthal orientation relative to the fault, even though they are the same distance away. This is interesting to observe, but since the GMPE curve ignores orientation it seems like it might also be interesting to try to average over different orientations for each distance. This could be facilitated if a number of observation points were placed at each distance, for a discrete set of distances, i.e., place the observation points on concentric rings with fixed RJB. It also seems like a much larger set of observation points could be used than 56, since the PC model is so quick to evaluate. If many points were placed on many different concentric rings, then one could average over all points at a given distance to get points that might be expected to agree better with the GMPE curve in Figure 6. It would then also be possible to explore in more detail how the PGV varies with orientation along each ring.

**Reply:** This is a very good observation. We thought about this already: variations in PGV at a given distance are likely due to radiation-pattern effects, in particular directivity. As pointed out by the reviewer, one could now do many more detailed tests, including using the PC approach to explore the ground motion dependency on azimuthal orientation. However, it would require the construction and validation of additional PC representations for a large number of observation stations, which are beyond the scope of this study.

Instead we refer to recently published study by Vyas et al (2016) that exactly addresses this question in great detail, with a range of simulations and 3000 randomly distributed sites.

The following sentences have been added to the discussion of Fig. 6 in our revised manuscript. "It is noted that two stations with similar Rjb distance can have very different PGV values. This is likely due to radiation-pattern effects, in particular directivity, which is addressed in great details by Vyas et al (2016)."

• In Figure 2 there are sets of points that have different colors/symbols that are arranged somewhat in rings, but the distance for each color do not seem to be constant. The use of colors/symbols is not explained anywhere I could find, and should be.

**Reply:** We have updated Figure 2 to provide details concerning the grouping of observation stations into four sets, and to indicate that the color/symbols are used to highlight this grouping. In addition, we also indicate the locations of two selected stations in Figure 4 and 5.

• Page 5, Table 2: The caption says that "(\*) denotes dependent parameters". It is not clear what this means. Does this refer to the comment in line 5 of this page, where it is noted that "These restrictions lead to nonlinear dependency between feasible ranges of different physical parameters"?

**Reply:** In the revised manuscript, we have modified the caption of Table 2 as follows: "Parameters governing fault plane configurations, (\*) denotes parameters whose feasible ranges are dependent on others."

• The fact that some of these parameters are constrained based on the choice of other parameters means that the probability distribution of parameters is not really given by (1) on page 5 as is stated. Some choices from this 7-dimensional box have probability zero due to the constraints, while others have greater probability due to several non-allowed choices mapping to the same set of modified parameters when the asperity falls near the edge of the fault plane. Does this affect the validity of the PC expansion and/or results? At any rate, this should be discussed.

**Reply:** A brief discussion has been added in the revised manuscript (beginning of section 3) in order to highlight the distinction between canonical random variables, which are iid uniform over the 7-dimensional hypercube, and physical parameters whose ranges may be interdependent. The PC expansion is constructed in terms of the canonical random variables, and its validity is tested using cross-validation and empirical error estimates.

• Page 17, Section 4.5: In this section it is stated that a uniform distribution of parameters over the 7-dimensional space ignores various geophysical constraints suggested by previous work. This is discussed in the context of choosing a prior for the Bayesian inference, but it seems like it would be even more important to incorporate these constraints into the analysis of Section 3, where the PC expansion is used to generate statistics on the PGV for comparison with the GMPE. Why should the statistics obtained with the uniform distribution be expected to match the GMPE well if it is known that this is the wrong distribution? This is addressed to some extent in Section 4.5 where the inversion that incorporates these constraints is then used to generate statistics that are compared to the GMPE curve in Figure 15. But at this point the inversion process has been used to to further constrain the posterior distribution based on trying to match the GMPE curve, so comparing the result to the GMPE curve does not seem to provide any validation that the PC expansion could predict the GMPE curve for other scenarios, for example. I may be missing the point here, but I think it needs more explanation.

1. "Why should the statistics obtained with the uniform distribution be expected to match the GMPE well if it is known that this is the wrong distribution?"

**Reply:** The PC statistics and GMPE results were compared to ensure that the model predictions describe a similar range, which consequently enables us to use the GMPE results as "observation data" for the purpose of parameter inference. Without this, it wouldn't be reasonable to use GMPE reference curve as ``observation'' in the Bayesian framework.

2. "so comparing the result to the GMPE curve does not seem to provide any validation that the PC expansion could predict the GMPE curve for other scenarios"

**Reply:** As pointed out earlier, the PC expansion was designed to provide an efficient representation of the rupture model behavior. In building the PC representation, we relied on uninformative prior, that spans a wide range of feasible scenarios. In Section 3, we verified the capability of the PC surrogate in reproducing the rupture model predictions over the considered parameter ranges. As discussed in Alexanderian et al. (2012), one of the advantages of having a suitable representation over a wide range of parameters is that the restriction of parameter ranges can be efficiently performed a posteriori, namely without the need of performing new model simulations. This advantage was specifically exploited in section 4.

As suggested by the referee, additional explanation has been incorporated in the revised manuscript concerning the construction and validation of the PC expansion, and later on the restrictions explored in the Bayesian analysis.

**Technical Corrections**

• Page 3, line 2: Presumably the rake is fixed at 0 degrees for a strike-slip event, but this should perhaps be stated?

**Reply:** As pointed out by the referee, the rake value has been added in the revised manuscript.

• Page 7, line 27: What are the index sets Si and Ti? The sets are used in the summations of (7a) and (7b) respectively, but not really defined.

**Reply:** As suggested by the referee, Si and Ti have been specified in the revised manuscript.

Proper latex fonts for trig functions should be used in expressions such as (A1), e.g. a cos β rather than acosβ.

**Reply:** This comment has been incorporated.

**References**

[revised manuscript text omitted]
<del>(obtained from , which are the most frequent values among 100 realizations following the scaling relation</del>

- 5 in Wells and Coppersmith (1994)sample realizations following scaling relations (e.g. Wells and Coppersmith, 1994; Mai and Beroza, 2000 The top of the fault plane is located 2 km below the ground surface. Figure 1 shows an example configuration of the fault plane, in which the red star denotes the hypocenter and the ellipse is the asperity with Gaussian slip distribution inside. The maximum slip  $S_{max}$  is chosen such that the mean slip (over the entire fault plane) remains constant (0.71 m) when varying the ellipse size (which ensures that the moment magnitude remains constant,  $M_w = 6.5$ ). The slip between the elliptical patch boundary
- 10 and dashed rectangle (Figure 1) is set to be  $S_{max}/e$  (where e is the Euler's number), the minimum value at the patch boundary from the Gaussian slip distribution. The slip between the solid and dashed rectangles (the horizontal and vertical gaps are 5% of the length and width of the fault plane, respectively) is tapered to avoid non-physical slip patterns. The entire fault plane is discretized in along-strike and down-dip directions with grid size of 0.02 km. We use a regularized Yoffe function (Tinti et al., 2005) with a rise time Tr = 1.25 s following source-scaling relations (Somerville et al., 1999) and slip acceleration time
- 15  $t_{acc} = 0.225$  s, as suggested by Tinti et al. (2005). At each node of the discretized fault plane we assign Tr,  $t_{acc}$ , slip-rate in along-strike and down-dip directions, and rupture time. We consider a rupture speed of  $0.75V_s$  km/s in all source models.

PGVs at a virtual network of  $N_{obs} = 56$  stations (Figure 2) are calculated from synthetic seismograms of the two horizontal components of ground motion at each site for a large set of source rupture models. We use COMPSYN (Spudich and Xu, 2003),

Figure 2. A virtual network of  $N_{obs} = 56$  stations where PGV responses are reported by the source model. The solid black line at the center denotes the length and location of the fault plane. Note, the 56 stations are grouped into four sets (indicated by different colors/symbols) according to their Rjb distances (see details in section 4).

Table 1. Velocity model used in this study, modified from Boore et al. (1997).

| Depth (km) | $V_p$ (km/s) | $V_s$ (km/s) |
|------------|--------------|--------------|
| 0          | 2.4          | 1.5          |
| 0.5        | 4.4          | 2            |
| 1.5        | 5.3          | 2.7          |
| 2.5        | 5.5          | 2.9          |
| 4          | 5.7          | 3.3          |
| 8          | 6.1          | 3.5          |
| 14         | 6.8          | 3.9          |
| 16.6       | 7.1          | 4.1          |
| 27         | 8            | 4.6          |
| 350        | 8.2          | 4.65         |

**Table 2. Parameters governing fault plane configurations, (\*) denotes dependent parameters whose feasible ranges are dependent on others.**

| Index | Parameter                              | Physical Interpretation                                                |
|-------|----------------------------------------|------------------------------------------------------------------------|
| 1     | AR                                     | Area ratio, $AR = \frac{\pi a b}{L * W} \in [0.05, 0.29]$              |
| 2     | $x_h (km)$                             | x-coordinate of the hypocenter $x_h \in [-13.5, 13.5]$                 |
| 3     | $z_h (km)$                             | z-coordinate of the hypocenter $\frac{y_h \in [-5,5]}{z_h \in [-5,5]}$ |
| 4     | $a\left( ^{st } ight) \left( km ight)$ | Semi-major axis $a \in [\sqrt{rac{
[revised manuscript text omitted]
 Ti) involving any of the random variables in i (including cross polynomials between variables in i and its complement i~, i ∪ i~ = {1,2,...,nd}). Note that by definition the two polynomial index sets satisfy Si ⊂ Ti.

**3.1 Validation of PC Models**

We first validate our PC surrogate models for PGVs at all stations. To this end, we introduce a second independent source model simulation ensemble (again an 8000 member LHS set  $\mathcal{P}_{LHS}^{valid} \subset \Xi$ ) for the purpose of validation. (Note that  $\mathcal{P}_{LHS}^{valid}$  is independent of the training set  $\mathcal{P}_{LHS}$ ). The following relative  $l_2$  error is then examined for PGVs at each station.

$$\epsilon_j = \sqrt{\frac{\sum_{k=1}^{N_{LHS}} (\tilde{\mathcal{Q}}_j(\boldsymbol{\xi}_k) - \mathcal{Q}_j(\boldsymbol{\xi}_k))^2}{\sum_{k=1}^{N_{LHS}} \mathcal{Q}_j(\boldsymbol{\xi}_k)^2}}, \ \forall j \in \{1, 2, \dots, N_{obs}\},\tag{9}$$

where  $\tilde{\mathcal{Q}}_j(\boldsymbol{\xi}_k)$  and  $\mathcal{Q}_j(\boldsymbol{\xi}_k)$  denote PC and source model responses, respectively, to  $\boldsymbol{\xi}_k$  at the *j*-th station.  $\boldsymbol{\xi}_k \in \mathcal{P}_{LHS}$  or  $\boldsymbol{\xi}_k \in \mathcal{P}_{LHS}^{valid}$  depending on the sample set used to estimate the errors.

Figure 3 shows relative error estimates of PC surrogate models over the training set ( $\mathcal{P}_{LHS}$ , blue dots) and the validation 25 set ( $\mathcal{P}_{LHS}^{valid}$ , red dots). It is not surprising to see slightly larger error estimates on the validation set, as the PC reconstruction

<sup>3The corresponding source code is available at https://github.com/mpf/spgl1